# Reward-free World Models for Online Imitation Learning

**Shangzhe Li**[1]  **Zhiao Huang**[2]  **Hao Su**[2][3]

## Abstract

Imitation learning (IL) enables agents to acquire skills directly from expert demonstrations, providing a compelling alternative to reinforcement learning. However, prior online IL approaches struggle with complex tasks characterized by high-dimensional inputs and complex dynamics. In this work, we propose a novel approach to online imitation learning that leverages reward-free world models. Our method learns environmental dynamics entirely in latent spaces without reconstruction, enabling efficient and accurate modeling. We adopt the inverse soft-Q learning objective, reformulating the optimization process in the Q-policy space to mitigate the instability associated with traditional optimization in the reward-policy space. By employing a learned latent dynamics model and planning for control, our approach consistently achieves stable, expert-level performance in tasks with high-dimensional observation or action spaces and intricate dynamics. We evaluate our method on a diverse set of benchmarks, including DMControl, MyoSuite, and ManiSkill2, demonstrating superior empirical performance compared to existing approaches.

## 1. Introduction

Imitation learning (IL) has garnered considerable attention due to its broad applications across various domains, such as robotic manipulation (Zhu et al., 2023; Chi et al., 2023) and autonomous driving (Hu et al., 2022; Zhou et al., 2021). Unlike reinforcement learning, where agents learn through reward signals, IL involves learning directly from expert demonstrations. Recent advances in offline IL, including Diffusion Policy (Chi et al., 2023) and Implicit BC (Florence et al., 2022), highlight the advantages of leveraging large datasets in conjunction with relatively straightforward

behavioral cloning (BC) methodologies. However, despite its wide applicability, IL methods that do not incorporate online interaction often suffer from poor generalization outside the expert data distribution, especially when encountering out-of-distribution states. Such limitations make these methods vulnerable to failure, as even minor perturbations in state can lead to significant performance degradation. This is often reflected in issues such as bias accumulation and suboptimal results (Reddy et al., 2019). These challenges stem from BC's inability to fully capture the underlying dynamics of the environment and its inherent lack of exploration capabilities (Garg et al., 2021).

To address these shortcomings, methods like GAIL (Ho & Ermon, 2016), SQIL (Reddy et al., 2019), IQ-Learn (Garg et al., 2021), and CFIL (Freund et al., 2023) have introduced value or reward estimation to facilitate a deeper understanding of the environment, while leveraging online interactions to enhance exploration. Nevertheless, these approaches continue to face substantial challenges, particularly when applied to tasks with high-dimensional observation and action spaces or complex dynamics. Additionally, framing online IL as a min-max optimization problem within the reward-policy space, often inspired by inverse reinforcement learning (IRL) techniques, introduces instability during training (Garg et al., 2021). Recent advancements in world models have demonstrated exceptional performance across a wide range of control tasks, underscoring their potential in complex decision-making and planning scenarios (Hafner et al., 2019a;b; 2020; 2023; Hansen et al., 2022; 2023). Specifically, world models offer advantages over model-free agents in terms of sampling complexity and future planning capabilities, resulting in superior performance on complex tasks (Hansen et al., 2022; 2023; Hafner et al., 2019a). Notably, decoder-free world models, which operate exclusively in latent spaces without reconstruction, have proven to be highly effective and efficient in modeling complex environment dynamics (Hansen et al., 2022; 2023).

Motivated by these insights, we explore the application of world models in the context of online imitation learning without rewards, enabling IL agents to develop a deeper understanding of environmental dynamics and improve their performance in tasks characterized by high-dimensional observations and complex dynamics. In this work, we present a novel approach to online imitation learning that leverages

---

[1]South China University of Technology [2]Hillbot Inc. [3]University of California, San Diego. Correspondence to: Shangzhe Li <shangzhe@unc.edu>.

*Proceedings of the 42nd International Conference on Machine Learning*, Vancouver, Canada. PMLR 267, 2025. Copyright 2025 by the author(s).

the strengths of decoder-free world models, specifically designed for complex tasks involving high-dimensional observations, intricate dynamics, and vision-based inputs. In contrast to conventional latent world models, which rely on reward and Q-function estimation, our approach completely eliminates the need for explicit reward modeling. We propose a framework for *reward-free* world models that redefines the optimization process within the Q-policy space, addressing the instability associated with min-max optimization in the reward-policy space. By utilizing an inverse soft-Q learning objective for the critic network (Garg et al., 2021), our method derives rewards directly from Q-values and the policy, effectively rendering the world model *reward-free*. Moreover, by performing imitation learning online, our model addresses key challenges in IL, such as out-of-distribution errors and bias accumulation.

Through online training with finite-horizon planning based on learned latent dynamics, our method demonstrates strong performance in complex environments. We evaluate our approach across a diverse set of locomotion and manipulation tasks, utilizing benchmarks from DMControl (Tunyasuvunakool et al., 2020), MyoSuite (Caggiano et al., 2022), and ManiSkill2 (Gu et al., 2023), and demonstrate superior empirical performance compared to existing online imitation learning methods.

Our contributions are as follows:

- We introduce a novel, robust methodology that leverages world models for online imitation learning, effectively addressing the challenges posed by complex robotics tasks.

- We propose an innovative gradient-free planning process, operating without explicit reward modeling, within the context of model predictive control.

- We showcase the model's effectiveness in inverse reinforcement learning tasks by demonstrating a positive correlation between decoded and ground-truth rewards.

## 2. Related Works

Our work builds upon literature in Imitation Learning (IL) and Model-based Reinforcement Learning.

**Imitation Learning**    Recent works regarding IL leveraged deep neural architectures to achieve better performance. Generative Adversarial Imitation Learning (GAIL) (Ho & Ermon, 2016) formulated the reward learning as a min-max problem similar to GAN (Goodfellow et al., 2014). Model-based Adversarial Imitation Learning (MAIL) (Baram et al., 2016) extended the GAIL approach to incorporate a forward model trained by data-driven methodology. Inverse

Soft Q-Learning (Garg et al., 2021) reformulated the learning objective of GAIL and integrated their findings into soft actor-critic (Haarnoja et al., 2018) and soft Q-learning agents for imitation learning. CFIL (Freund et al., 2023) introduced a coupled flow approach for reward generation and policy learning using expert demonstrations. ValueDICE (Kostrikov et al., 2019) proposed an off-policy imitation learning approach by transforming the distribution ratio estimation objective. (Das et al., 2021) proposed a model-based inverse RL approach by predicting key points for imitation learning tasks. SQIL (Reddy et al., 2019) proposed an online imitation learning algorithm with soft Q functions. Diffusion Policy (Chi et al., 2023) is a recent offline IL method using a diffusion model for behavioral cloning. Implicit BC (Florence et al., 2022) discovers that treating supervised policy learning with an implicit model generally improves the empirical performance for robot learning tasks. Hybrid inverse reinforcement learning (Ren et al., 2024) proposed a new methodology leveraging a mixture of online and expert demonstrations for agent training, achieving robust performance in environments with stochasticity. Prior works (Englert et al., 2013; Hu et al., 2022; Igl et al., 2022) explored the potentials of model-based imitation learning on real-world robotics control and autonomous driving. EfficientImitate (Yin et al., 2022) combined EfficientZero (Ye et al., 2021) with adversarial imitation learning, achieving excellent results in DMControl (Tassa et al., 2018) imitation learning tasks. V-MAIL (Rafailov et al., 2021) introduced a model-based approach for imitation learning using variational models. CMIL (Kolev et al., 2024) proposed an imitation learning approach with conservative world models for image-based manipulation tasks. Ditto (DeMoss et al., 2023) developed an offline imitation learning approach with Dreamer V2 (Hafner et al., 2020) and adversarial imitation learning. DMIL (Zhang et al., 2023) utilized a discriminator to simultaneously evaluate both the accuracy of the dynamics and the suboptimality of model rollout data relative to real expert demonstrations in the context of offline imitation learning.

**Model-based Reinforcement Learning**    Contemporary model-based RL methods often learn a dynamics model for future state prediction via data-driven approaches. PlaNet (Hafner et al., 2019b) was introduced as a model-based learning approach for partially observed MDPs by proposing a recurrent state-space model (RSSM) and an evidence lower-bound (ELBO) training objective. Dreamer algorithm (Hafner et al., 2019a; 2020; 2023) is a model-based reinforcement learning approach that uses a learned world model to efficiently simulate future trajectories in a latent space, allowing an agent to learn and plan effectively. TD-MPC series (Hansen et al., 2022)(Hansen et al., 2023) learns a scalable world model for model predictive control using temporal difference learning objective.

Our approach employs a model-based methodology to address challenges in online imitation learning. By integrating a data-driven approach for latent dynamics learning with planning for control, the agent is able to effectively capture and leverage the underlying environment dynamics. Empirical evaluations demonstrate that our model achieves superior performance on complex online imitation learning tasks compared to existing methods.

## 3. Preliminary

We model the decision-making process in the environment as a Markov Decision Process (MDP), which can be defined as a tuple $\langle \mathcal{S}, \mathcal{A}, p_0, \mathcal{P}, r, \gamma \rangle$. $\mathcal{S}$ and $\mathcal{A}$ represent state and action space. $p_0$ is the initial state distribution and $\mathcal{P} : \mathcal{S} \times \mathcal{A} \to \Delta_{\mathcal{S}}$ is the transition probability. $r(\mathbf{s}, \mathbf{a}) \in \mathcal{R}$ is the reward function and $\mathcal{R}$ is the reward space. $\gamma \in (0, 1)$ is the discount factor. We denote the expert state-action distribution as $\rho_E$ and the behavioral distribution as $\rho_\pi$. Similarly, we denote the expert policy as $\pi_E$ and the behavioral policy as $\pi$. $\Pi$ is the set of all stochastic stationary policies that sample an action $\mathbf{a} \in \mathcal{A}$ given a state $\mathbf{s} \in \mathcal{S}$. $\mathcal{Z}$ is the space for the latent representation of the original state observations, and $\mathcal{Q}$ is the space for all possible Q functions. $\mathcal{H}(\cdot)$ represents the entropy of a distribution.

**Maximum Entropy Inverse Reinforcement Learning** Inverse Reinforcement Learning (IRL) focuses on recovering a specific reward function $r(s, a)$ in the reward space $\mathcal{R}$ given a certain amount of expert samples using expert policy $\pi_E$. Maximum entropy IRL (Ziebart et al., 2008) seeks to solve this problem by optimizing $\max_{r \in \mathcal{R}} \min_{\pi \in \Pi} \mathbb{E}_{\rho_E}[r(\mathbf{s}, \mathbf{a})] - (\mathbb{E}_{\rho_\pi}[r(\mathbf{s}, \mathbf{a})] + \mathcal{H}(\pi))$. GAIL (Ho & Ermon, 2016) generalized the objective into a form including an explicit reward mapping with a convex regularizer $\psi(r)$:

$$\max_{r \in \mathcal{R}} \min_{\pi \in \Pi} \quad \mathbb{E}_{\rho_E}[r(\mathbf{s}, \mathbf{a})] - \mathbb{E}_{\rho_\pi}[r(\mathbf{s}, \mathbf{a})] - \mathcal{H}(\pi) - \psi(r) \quad (1)$$

For a non-restrictive set of reward functions $\mathcal{R} = \mathbb{R}^{\mathcal{S} \times \mathcal{A}}$, the objective can be reformulated into a minimization of the statistical distance between distributions $\rho_E$ and $\rho_\pi$ (Ho & Ermon, 2016):

$$\min_\pi \quad d_\psi(\rho_\pi, \rho_E) - \mathcal{H}(\pi) \quad (2)$$

**Inverse Soft-Q Learning** Prior work (Garg et al., 2021) introduced a bijection mapping $\mathcal{T}^\pi : \mathbb{R}^{\mathcal{S} \times \mathcal{A}} \to \mathbb{R}^{\mathcal{S} \times \mathcal{A}}$ between Q space $\mathcal{Q}$ and reward space $\mathcal{R}$, i.e., the inverse Bellman operator:

$$(\mathcal{T}^\pi Q)(\mathbf{s}, \mathbf{a}) = Q(\mathbf{s}, \mathbf{a}) - \gamma \mathbb{E}_{\mathbf{s}' \sim \mathcal{P}(\cdot|\mathbf{s}, \mathbf{a})} V^\pi(\mathbf{s}') \quad (3)$$

where $V^\pi(\mathbf{s}) = \mathbb{E}_{\mathbf{a} \sim \pi(\cdot|\mathbf{s})}[Q(\mathbf{s}, \mathbf{a}) - \log \pi(\mathbf{a}|\mathbf{s})]$. The reward decoding is defined as $r = \mathcal{T}^\pi Q$. By applying the

operator $\mathcal{T}^\pi$ over Eq.1, prior work reformulated the GAIL training objective in Q-policy space (Garg et al., 2021):

$$\mathcal{J}(\pi, Q) = \mathbb{E}_{(\mathbf{s}, \mathbf{a}) \sim \rho_E} \left[ Q(\mathbf{s}, \mathbf{a}) - \gamma \mathbb{E}_{\mathbf{s}' \sim \mathcal{P}(\cdot|\mathbf{s}, \mathbf{a})} V^\pi(\mathbf{s}') \right]$$
$$- \mathbb{E}_{(\mathbf{s}, \mathbf{a}) \sim \rho_\pi} \left[ V^\pi(\mathbf{s}) - \gamma \mathbb{E}_{\mathbf{s}' \sim \mathcal{P}(\cdot|\mathbf{s}, \mathbf{a})} V^\pi(\mathbf{s}') \right]$$
$$- \psi(\mathcal{T}^\pi Q)$$
$$(4)$$

which is the inverse soft-Q objective for critic learning. In this way, we can perform imitation learning by leveraging actor-critic architecture. The critic and policy can be learned by finding the saddle point in a joint optimization problem $Q^* = \text{argmax}_{Q \in \mathcal{Q}} \min_{\pi \in \Pi} \mathcal{J}(\pi, Q)$ and $\pi^* = \text{argmin}_{\pi \in \Pi} \max_{Q \in \mathcal{Q}} \mathcal{J}(\pi, Q)$. (Garg et al., 2021) proved the uniqueness of the saddle point. For a fixed Q, the optimization for policy has a closed-form solution, which is the softmax policy:

$$\pi_Q(\mathbf{a}|\mathbf{s}) = \frac{\exp Q(\mathbf{s}, \mathbf{a})}{\sum_\mathbf{a} \exp Q(\mathbf{s}, \mathbf{a})} \quad (5)$$

In the actor-critic setting, we can optimize the policy $\pi$ using maximum-entropy RL objective, which approximates $\pi_Q$, and learn critic using:

$$\max_{Q \in \mathcal{Q}} \mathcal{J}(\pi, Q)$$
$$= \max_{Q \in \mathcal{Q}} \left[ \mathbb{E}_{(\mathbf{s}, \mathbf{a}) \sim \rho_E} \left[ \phi(Q(\mathbf{s}, \mathbf{a}) - \gamma \mathbb{E}_{\mathbf{s}' \sim \mathcal{P}(\cdot|\mathbf{s}, \mathbf{a})} V^\pi(\mathbf{s}')) \right] \right.$$
$$\left. - \mathbb{E}_{(\mathbf{s}, \mathbf{a}) \sim \rho_\pi} \left[ V^\pi(\mathbf{s}) - \gamma \mathbb{E}_{\mathbf{s}' \sim \mathcal{P}(\cdot|\mathbf{s}, \mathbf{a})} V^\pi(\mathbf{s}') \right] \right]$$
$$(6)$$

where $\phi$ is a concave function. Specifically, if we leverage $\chi^2$ regularization, we will have $\phi(x) = x - \frac{1}{4\alpha}x^2$. The scalar coefficient $\alpha$ controls the strength of $\chi^2$ regularization in the inverse soft-Q objective. Intuitively, the additonal regularization term penalizes the magnitude of the estimated reward. Prior work (Al-Hafez et al., 2023) interpreted the objective with this regularizer as minimizing the squared Bellman error, establishing a connection between inverse soft-Q learning and SQIL (Reddy et al., 2019). A detailed empirical analysis on hyperparameter $\alpha$ is shown in Appendix E.3.

## 4. Methodology

In reinforcement learning, world models typically regress explicit reward signals provided by the environment. In imitation learning, prior approaches (Kolev et al., 2024; DeMoss et al., 2023) aim to train a reward model through adversarial objectives alongside a separate critic network trained on temporal difference objectives. In contrast, we

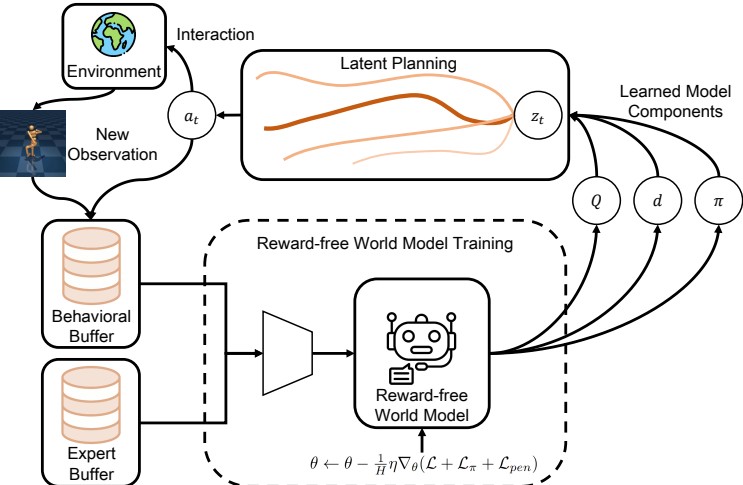

*Figure 1.* **IQ-MPC** We demonstrate the training workflow for IQ-MPC. The reward-free world model leverages both expert and behavioral data for training, using objectives in Section 4.1. The policy prior from the world model guides the MPPI planning process along with rewards decoded from Q estimations. The detailed planning process is revealed in Algorithm 1.

eliminate the need for a separate reward model by retrieving rewards directly from the learned critic. To this end, we propose a *reward-free* world model that learns exclusively from reward-free expert demonstrations and environment interactions, without training a dedicated reward model. Furthermore, since our model can decode dense rewards from the critic, it can solve inverse reinforcement learning tasks using reward-free interactions and a limited set of expert demonstrations that include only states and actions.

### 4.1. Learning Process of a Reward-free World Model

World models used in reinforcement learning settings often contain a reward model $R(\mathbf{z}, \mathbf{a})$ that requires supervised learning using explicit reward signals from the online environment interactions or the offline data. However, if our learning objective is able to form a bijection between Q space $\mathcal{Q}$ and reward space $\mathcal{R}$, it would be natural to decode the reward from the Q value instead of learning another separate mapping for the reward, which also enables the world model to perform imitation learning with expert demonstrations without explicit reward signal. An overview of our proposed method is shown in Figure 1. The detailed training algorithm is shown in Algorithm 2. We also provide a theoretical analysis of our training objective, as detailed in Section 4.2 Appendix H.3.

**Model Components** We introduce our approach for imitation learning as **I**nverse Soft-**Q** Learning for **M**odel **P**redictive **C**ontrol, or **IQ-MPC** as an abbreviation. Our architecture consists of four components:

$$\text{Encoder:} \quad \mathbf{z} = h(\mathbf{s}) \tag{7}$$
$$\text{Latent dynamics:} \quad \mathbf{z}' = d(\mathbf{z}, \mathbf{a}) \tag{8}$$

$$\text{Value function:} \quad \hat{q} = Q(\mathbf{z}, \mathbf{a}) \tag{9}$$
$$\text{Policy prior:} \quad \hat{\mathbf{a}} = \pi(\mathbf{z}) \tag{10}$$

where $\mathbf{s}$ and $\mathbf{a}$ are states and actions, $\mathbf{z}$ is latent representations. The policy prior $\pi$ guides the model predictive planning process, along with rewards decoded from the value function $Q$. We maintain two separate replay buffers $\mathcal{B}_E$ and $\mathcal{B}_\pi$ for expert and behavioral data storage respectively. Behavioral data are collected during the learning process. For simplicity, we denote the sampling process from the joint buffer as $\mathcal{B} = \mathcal{B}_E \cup \mathcal{B}_\pi$. We sample trajectories with short horizons of length $H$ from the replay buffers.

**Model Learning** We learn the encoder $h$, latent dynamics $d(\mathbf{z}, \mathbf{a})$ and Q function $Q(\mathbf{z}, \mathbf{a})$ jointly by minimizing the objective for prediction consistency and critic learning:

$$\mathcal{L} = \sum_{t=0}^{H} \lambda^t \left( \mathbb{E}_{(\mathbf{s}_t, \mathbf{a}_t, \mathbf{s}_t') \sim \mathcal{B}} \|\mathbf{z}_{t+1} - \text{sg}(h(\mathbf{s}_t'))\|_2^2 \right) + \mathcal{L}_{iq} \tag{11}$$

where sg is the stop gradient operator and $\mathcal{L}_{iq}$ is the inverse soft-Q critic objective, which is a modification for horizon $H$ and latent representation $\mathbf{z}$ from Eq.7 based on Eq.6:

$$\mathcal{L}_{iq}(Q, \pi)$$
$$= \sum_{t=0}^{H} \lambda^t \left[ - \mathbb{E}_{(\mathbf{s}_t, \mathbf{a}_t, \mathbf{s}_t') \sim \mathcal{B}_E} \left[ Q(\mathbf{z}_t, \mathbf{a}_t) - \gamma \bar{V}^\pi(h(\mathbf{s}_t')) \right] \right.$$
$$+ \mathbb{E}_{\mathbf{s}_0 \sim \mathcal{B}_E} \left[ (1 - \gamma) V^\pi(\mathbf{z}_0) \right]$$
$$\left. + \mathbb{E}_{(\mathbf{s}_t, \mathbf{a}_t, \mathbf{s}_t') \sim \mathcal{B}} \frac{1}{4\alpha} \left[ Q(\mathbf{z}_t, \mathbf{a}_t) - \gamma \bar{V}^\pi(h(\mathbf{s}_t')) \right]^2 \right] \tag{12}$$

Compared to Eq.6, the key difference is the second term of the objective, which computes the original value difference $\mathbb{E}_{(\mathbf{s}_t,\mathbf{a}_t,\mathbf{s}'_t)\sim\mathcal{B}_\pi}[V^\pi(\mathbf{z}_t) - \gamma V^\pi(\mathbf{z}'_t)]$ using only the representation of the initial state $\mathbf{s}_0$. This reformulation, derived in Lemma H.3 (Appendix H.1), yields more stable Q estimation, as confirmed by the ablation in Appendix E.3. We also apply $\chi^2$ regularization, as noted in (Garg et al., 2021). We leverage $\lambda \in (0,1]$ as a constant discounting weight over the horizon, guaranteeing the influence to be smaller for states and actions farther ahead. Note that $\lambda$ here differs from the environment discount factor $\gamma$. All value functions in the objective are computed from Q and policy network via $V^\pi(\mathbf{z}) = \mathbb{E}_{\mathbf{a}\sim\pi(\cdot|\mathbf{z})}[Q(\mathbf{z},\mathbf{a}) - \beta\log\pi(\mathbf{a}|\mathbf{z})]$, where $\beta$ is the entropy coefficient. We will further discuss the selection of $\beta$ in the policy learning part. Especially, $\bar{V}^\pi(h(\mathbf{s}'))$ is the value function computed by the target Q network $\bar{Q}$. $\mathbf{z}$ is retrieved by rolling out dynamics model from the latent representation of the first state:

$$\mathbf{z}_{t+1} = d(\mathbf{z}_t,\mathbf{a}_t), \quad \mathbf{z}_0 = h(\mathbf{s}_0)$$

We update the Q, encoder, and dynamics network by minimizing Eq.11 while keeping policy prior $\pi$ fixed.

**Policy Prior Learning** We choose to learn the policy prior network with the maximum entropy reinforcement learning. We minimize the following maximum entropy RL objective using data sampled from both the expert buffer and the behavioral buffer:

$$\mathcal{L}_\pi = \sum_{t=0}^{H}\lambda^t\left[\mathbb{E}_{(\mathbf{s}_t,\mathbf{a}_t)\sim\mathcal{B}}\left[-Q(\mathbf{z}_t,\pi(\mathbf{z}_t))+\beta\log(\pi(\cdot|\mathbf{z}_t))\right]\right] \tag{13}$$

$\beta$ is an entropy coefficient which is a fixed scalar. (Hansen et al., 2023) experimented on adaptive entropy coefficient and observed no performance improvement on model predictive control compared to a fixed scalar. Therefore, we also choose not to leverage a learnable $\beta$ for simplicity. We prove in Theorem H.4 that this policy update can achieve $\pi^* = \mathrm{argmax}_{\pi\in\Pi}\min_{Q\in\mathcal{Q}}\mathcal{L}_{iq}(Q,\pi)$ to find the saddle point.

**Balancing Critic and Policy Training** We observe unstable training processes in some tasks due to the imbalance between the critic and the policy. When the discriminative power of the critic is too strong, the policy prior $\pi$ may fail to learn properly. In those cases the Q value difference between expert batch and behavioral batch $\mathbb{E}_{(\mathbf{s},\mathbf{a})_{(0:H)}\sim\mathcal{B}_E}Q(\mathbf{z}_t,\mathbf{a}_t) - \mathbb{E}_{(\mathbf{s},\mathbf{a})_{(0:H)}\sim\mathcal{B}_\pi}Q(\mathbf{z}_t,\mathbf{a}_t)$ will not converge. To mitigate this issue, we choose to use the Wasserstein-1 metric for gradient penalty (Gulrajani et al., 2017; Garg et al., 2021) in addition to the original inverse soft-Q objective, enforcing Lipschitz condition for the gra-

---

**Algorithm 1** IQ-MPC (*inference*)

**Require:** $\theta$ : learned network parameters
      $\mu^0,\sigma^0$: initial parameters for $\mathcal{N}$
      $N, N_\pi$: number of sample/policy trajectories
      $\mathbf{s}_t, H$: current state, rollout horizon
1: Encode state $\mathbf{z}_t \leftarrow h_\theta(\mathbf{s}_t)$
2: **for** each iteration $j = 1..J$ **do**
3:   Sample $N$ trajectories of length $H$ from $\mathcal{N}(\mu^{j-1},(\sigma^{j-1})^2\mathbf{I})$
4:   Sample $N_\pi$ trajectories of length $H$ using $\pi_\theta, d_\theta$
    *// Estimate trajectory returns $\phi_\Gamma$ using $d_\theta, Q_\theta, \pi_\theta$,*
    *starting from $\mathbf{z}_t$ and initialize $\phi_\Gamma = 0$:*
5:   **for** all $N + N_\pi$ trajectories $(\mathbf{a}_t,\mathbf{a}_{t+1},\ldots,\mathbf{a}_{t+H})$ **do**
6:     **for** step $t = 0..H-1$ **do**
7:       $\mathbf{z}_{t+1} \leftarrow d_\theta(\mathbf{z}_t,\mathbf{a}_t)$   ◁ *Latent transition*
8:       $\hat{\mathbf{a}}_{t+1} \sim \pi_\theta(\cdot|\mathbf{z}_{t+1})$
9:       Compute $V^\pi(\mathbf{z}_{t+1})$ via $Q_\theta(\mathbf{z}_{t+1},\hat{\mathbf{a}}_{t+1}) - \beta\log\pi_\theta(\hat{\mathbf{a}}_{t+1}|\mathbf{z}_{t+1})$
10:      $r(\mathbf{z}_t,\mathbf{a}_t) = Q_\theta(\mathbf{z}_t,\mathbf{a}_t) - \gamma V^\pi(\mathbf{z}_{t+1})$ ◁ *Reward decoding*
11:      $\phi_\Gamma = \phi_\Gamma + \gamma^t[r(\mathbf{z}_t,\mathbf{a}_t) - \beta\log\pi_\theta(\mathbf{a}_t|\mathbf{z}_t)]$
12:     **end for**
13:     $\phi_\Gamma = \phi_\Gamma + \gamma^H V^\pi(\mathbf{z}_H)$  ◁ *Terminal value*
14:   **end for**
15:   *// Update parameters $\mu,\sigma$ for next iteration:*
16:   $\mu^j,\sigma^j \leftarrow$ MPPI update with $\phi_\Gamma$.
17: **end for**
18: **return** $\mathbf{a} \sim \mathcal{N}(\mu^J,(\sigma^J)^2\mathbf{I})$

---

dient:

$$\mathcal{L}_{pen} = \sum_{t=0}^{H}\lambda^t\left[\mathbb{E}_{(\hat{\mathbf{s}}_t,\hat{\mathbf{a}}_t)\sim\mathcal{B}}\left(\|\nabla Q(\hat{\mathbf{z}}_t,\hat{\mathbf{a}}_t)\|_2 - 1\right)^2\right] \tag{14}$$

In Eq.14, $\hat{\mathbf{s}}$ and $\hat{\mathbf{a}}$ are sampled from the straight line between samples from expert buffer $(\mathbf{s},\mathbf{a}) \sim \mathcal{B}_E$ and behavioral buffer $(\mathbf{s},\mathbf{a}) \sim \mathcal{B}_\pi$ by linear interpolation. $\nabla$ is the gradient with respect to the interpolated input $\hat{\mathbf{z}}_t$ and $\hat{\mathbf{a}}_t$. By incorporating this additional objective, we can enforce unit gradient norm over the straight lines between state-action distribution $\rho_\pi$ and $\rho_E$. We show the ablation study regarding this regularization term in Appendix E.3.

### 4.2. Theoretical Analysis on the Learning Objective

In this section, we demonstrate theoretically that the learning objectives of IQ-MPC effectively minimize the value difference between the current policy and the expert, ensuring that Q-value estimation can follow as the latent dynamics model learns. We begin by utilizing the following lemma established in (Kolev et al., 2024):

**Lemma 4.1** (Bounded Suboptimality). *Given an unknown latent MDP $\mathcal{M}$ and our learned latent MDP $\hat{\mathcal{M}}$ with transition probabilities $d$ and $\hat{d}$ in the latent state space $\mathcal{Z}$ and*

*action space $\mathcal{A}$, and letting $R_{\max}$ denote the maximum reward of the unknown MDP, the difference between the expected return of the current policy, $\eta_{\mathcal{M}}^{\pi}$, and that of the expert policy, $\eta_{\mathcal{M}}^{\pi_E}$, is bounded by:*

$$|\eta_{\mathcal{M}}^{\pi_E} - \eta_{\mathcal{M}}^{\pi}| \leq \underbrace{\frac{2R_{\max}}{1-\gamma} D_{TV}(\rho_{\hat{\mathcal{M}}}^{\pi}, \rho_{\mathcal{M}}^{\pi_E})}_{T1}$$
$$+ \underbrace{\frac{\gamma R_{\max}}{(1-\gamma)^2} \mathbb{E}_{\rho_{\hat{\mathcal{M}}}^{\pi}} \left[ D_{TV}(d(\mathbf{z}'|\mathbf{z},\mathbf{a}), \hat{d}(\mathbf{z}'|\mathbf{z},\mathbf{a})) \right]}_{T2}$$

Our critic and policy objectives can be interpreted as a minmax optimization of Eq.4. Moreover, this approach can be viewed as minimizing a statistical distance with an entropy term, corresponding to Eq.2. Thus, on one hand, our critic and policy objectives effectively minimize T1 in the bound provided in Lemma 4.1. On the other hand, optimizing our consistency loss in Eq.11 is approximately minimizing the second term T2 in the bound. Our training objective ensures that as the dynamics model learns, it simultaneously minimizes the upper bound of the deviation between the expected return for current policy $\pi$ and the expert expected return. A more detailed analysis on T2 is given in Appendix H.3.

### 4.3. Planning with Policy Prior

Similar to TD-MPC (Hansen et al., 2022) and TD-MPC2 (Hansen et al., 2023), we utilize the Model Predictive Control (MPC) framework for local trajectory optimization over the latent representations and acquire control action by leveraging Model Predictive Path Integral (MPPI)(Williams et al., 2015) with sampled action sequences $(\mathbf{a}_t, \mathbf{a}_{t+1}, ..., \mathbf{a}_{t+H})$ of length $H$. Instead of planning with explicit reward models like TD-MPC and TD-MPC2, we estimate the parameters $(\mu^*, \sigma^*)$ using derivative-free optimization with reward information decoded from the critic's estimation:

$$\mu^*, \sigma^* = \underset{(\mu,\sigma)}{\arg\max} \, \mathbb{E}_{(\mathbf{a}_t,...,\mathbf{a}_{t+H}) \sim \mathcal{N}(\mu,\sigma^2)} \Big[ \gamma^H V^\pi(\mathbf{z}_{t+H})$$
$$+ \sum_{h=0}^{H-1} \gamma^h (r(\mathbf{z}_{t+h}, \mathbf{a}_{t+h}) - \beta \log \pi(\mathbf{a}_{t+h}|\mathbf{z}_{t+h})) \Big] \tag{15}$$

where $\mu, \sigma \in \mathbb{R}^{H \times m}, m = \dim \mathcal{A}$. $(\mathbf{z}_t, ..., \mathbf{z}_{t+H})$ are computed by unrolling with $(\mathbf{a}_t, .., \mathbf{a}_{t+H})$ using dynamics model $d_\theta$. The reward $r(\mathbf{z}, \mathbf{a})$ is computed by $Q(\mathbf{z}, \mathbf{a}) - \gamma \mathbb{E}_{\mathbf{z}' \sim d(\cdot|\mathbf{z},\mathbf{a})} V^\pi(\mathbf{z}')$. Eq.15 is solved by iteratively computing soft expected return $\phi_\Gamma$ of sampled actions from $\mathcal{N}(\mu, \sigma^2)$ and update $\mu, \sigma$ based on weighted average with $\phi_\Gamma$. We describe the detailed planning procedure in Algorithm 1. Eq.15 is an estimation of the soft-Q learning

objective (Haarnoja et al., 2017) for RL with horizon $H$. After iteration, we execute the first action sampled from the normal distribution $a_t \sim \mathcal{N}(\mu_t^*, (\sigma_t^*)^2 \mathrm{I})$ in the environment to collect a new trajectory for behavioral buffer $\mathcal{B}_\pi$.

## 5. Experiments

We conduct experiments for locomotion and manipulation tasks to demonstrate the effectiveness of our approach. We choose to leverage the online version of IQ-Learn+SAC (referred to as IQL+SAC in the experiment plots) (Garg et al., 2021), CFIL+SAC (Freund et al., 2023), and HyPE (Ren et al., 2024) as our baselines for comparison studies. The results presented below for our IQ-MPC model are obtained through planning. We provide an analysis of the computational overhead of our model in Appendix F. The empirical results regarding state-based and visual experiments are shown in Section 5.1. We experiment on the reward recovery capability of our IQ-MPC model, for which we reveal the results in Appendix G. We conduct ablation studies for our model, which are discussed in Section 5.2. The details of the environments and tasks can be found in Appendix D. We also analyze the training time and the robustness of our model under noisy environment dynamics. The corresponding results are presented in Appendix F and E.4.

### 5.1. Main Results

#### 5.1.1. STATE-BASED EXPERIMENTS

**Locomotion Tasks** We benchmark our algorithm on DM-Control (Tunyasuvunakool et al., 2020), evaluating tasks in both low- and high-dimensional environments. Our method outperforms baselines in performance and training stability. We use 100 expert trajectories for low-dimensional tasks (Hopper, Walker, Quadruped, Cheetah), 500 for Humanoid, and 1000 for Dog (both high-dimensional). Each trajectory contains 500 steps, sampled using trained TD-MPC2 world models (Hansen et al., 2023). Performance is averaged over 3 seeds per task. The results are demonstrated in Figure 2. Our method is comparable to HyPE in the Quadruped Run and Cheetah Run tasks, while outperforming all other baselines in the remaining tasks. We also conducted high-dimensional experiments on various tasks in the Dog environment, with results provided in Appendix E.1.

**Manipulation Tasks** We consider manipulation tasks with a dexterous hand from MyoSuite (Caggiano et al., 2022) to show the capability and robustness of our IQ-MPC model in high-dimensional and complex dynamics scenarios. We leverage 100 expert trajectories with 100 steps sampled from trained TD-MPC2 for each task. We evaluate the episode reward and success rate of our model along with

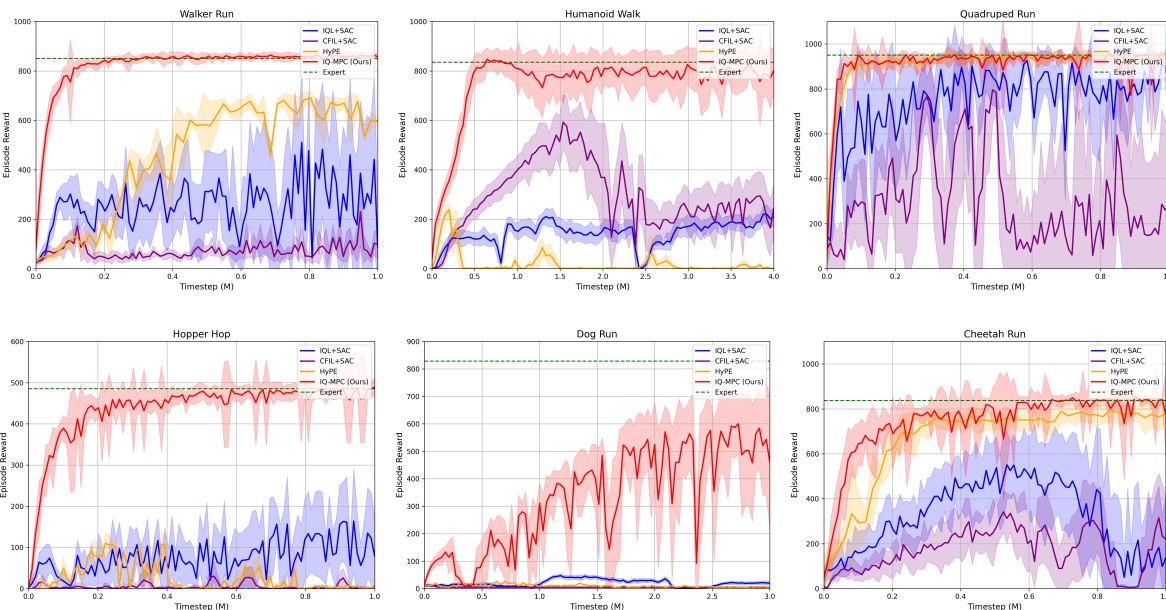

*Figure 2.* **Locomotion Results** Our method demonstrates much stabler performance near expert level compared to baseline methods. In the plots, blue lines refers to the online version of IQL+SAC (Garg et al., 2021), orange lines refers to the HyPE method (Ren et al., 2024), purple lines refers to the CFIL+SAC (Freund et al., 2023) baseline and red lines refers to our IQ-MPC model. The dotted green lines are the mean episode reward for the expert trajectories used during training.

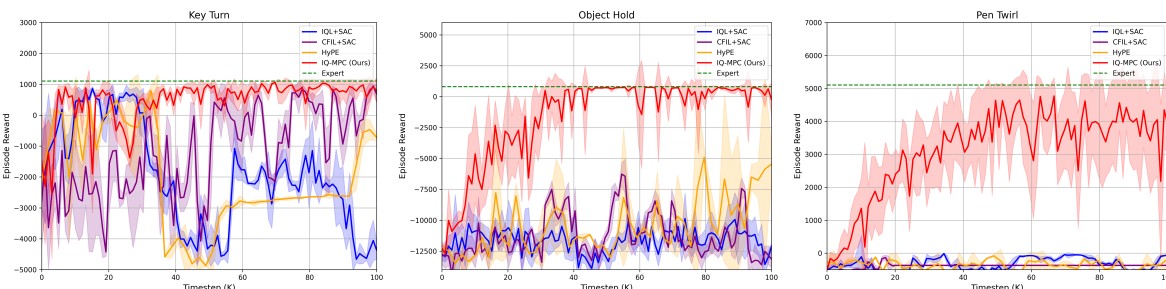

*Figure 3.* **Manipulation Results in MyoSuite** Our IQ-MPC shows stable and outperforming results in MyoSuite manipulation experiments with dexterous hands. In the plots, the color settings are the same as those in Figure 2. In the Pen Twirl task, the CFIL+SAC agent is unable to train after 20K time steps. Thus, we interpolate the rest of the time steps with a straight line in the plot.

IQ-Learn+SAC, HyPE, and CFIL+SAC. We show superior empirical performance in three different tasks, including object holding, pen twirling, and key turning. Regarding the results for episode reward, we refer to Figure 3. Table 1 shows the success rate results. We take the mean for 3 seeds regarding the performance for each task. We have conducted additional experiments on ManiSkill2 (Gu et al., 2023), for which we refer to Appendix E.2.

### 5.1.2. VISUAL EXPERIMENTS

We further investigate the capability of handling visual tasks for our IQ-MPC model. We conduct the experiments on locomotion tasks in DMControl with visual observations. We demonstrate that our IQ-MPC model can cope with visual modality inputs by only replacing the encoder with a shallow convolution network and keeping the rest of the model unchanged. We sample the expert data using trained TD-MPC2 models with visual inputs. We take 100 expert trajectories for each task. The expert trajectories contain actions and RGB frame observations. We leverage a modification of IQL+SAC as our baseline. We add the same convolutional encoder as our IQ-MPC for processing visual inputs and keep the rest of the architecture the same. We perform superior to the baseline model in a series of visual experiments in the DMControl environment. We demonstrate the results in Figure 4.

### 5.2. Ablation Studies

In this section, we will show the ablation studies of our model, including the ablation over expert trajectory num-

| Method | IQL+SAC | CFIL+SAC | HyPE | IQ-MPC(Ours) |
|---|---|---|---|---|
| Key Turn | 0.72±0.04 | 0.65±0.08 | 0.55 ± 0.09 | **0.87±0.03** |
| Object Hold | 0.00 ± 0.00 | 0.01±0.01 | 0.13 ± 0.10 | **0.96±0.03** |
| Pen Twirl | 0.00 ± 0.00 | 0.00±0.00 | 0.00 ± 0.00 | **0.73±0.05** |

*Table 1.* **Manipulation Success Rate Results in MyoSuite** We evaluate the success rate of IQ-MPC on the Key Turn, Object Hold, and Pen Twirl tasks in MyoSuite. Our IQ-MPC demonstrates strength in handling complex manipulation tasks with dexterous hands and musculoskeletal motor control. We show the results by averaging over 100 trajectories and evaluating over 3 random seeds.

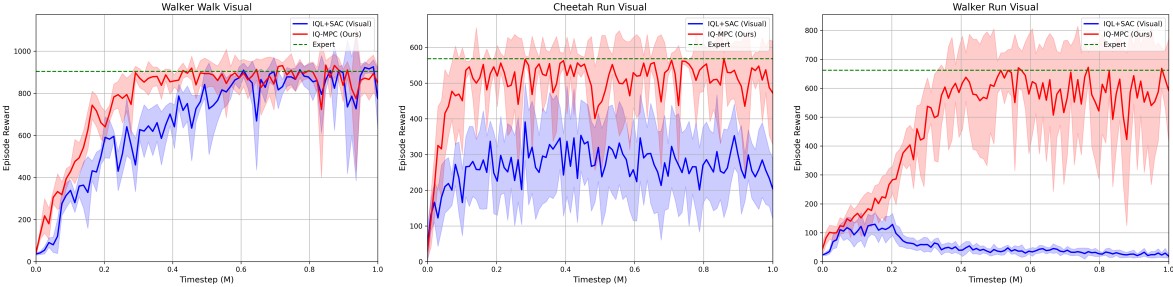

*Figure 4.* **Results for Visual Experiments** Our IQ-MPC (red lines) shows stable and expert-level results in visual observation tasks. In the plots, we denote the IQL+SAC with an additional convolutional encoder as IQL+SAC (Visual) (blue lines). Our model outperforms IQL+SAC (Visual) in the Cheetah Run and Walker Run, and it has comparable performance in the Walker Walk task. The expert trajectories used for training are sampled from TD-MPC2 trained on visual observations.

bers. Regarding the ablations for objective formulation, gradient penalty selection, and hyperparameter $\alpha$, we refer to Appendix E.3.

We ablate over the expert trajectories used for IQ-MPC training. We demonstrate our results with 100, 50, 10, and 5 expert trajectories in the Hopper Hop task and Object Hold task. We show that our world model can still reach expert-level performance with only a small amount of expert demonstrations but with slower convergence. The instability is observed with 5 expert trajectories in the Hopper Hop task. We reveal the empirical results for this ablation in Figure 5.

# 6. Conclusions and Broader Impact

We propose an online imitation learning approach that utilizes reward-free world models to address tasks in complex environments. By incorporating latent planning and dynamics learning, our model can have a deeper understanding of intricate environment dynamics. We demonstrate stable, expert-level performance on challenging tasks, including dexterous hand manipulation and high-dimensional locomotion control. In terms of broader impact, our model holds potential for real-world applications in manipulation and locomotion, particularly for tasks that involve visual inputs and complex environment dynamics.

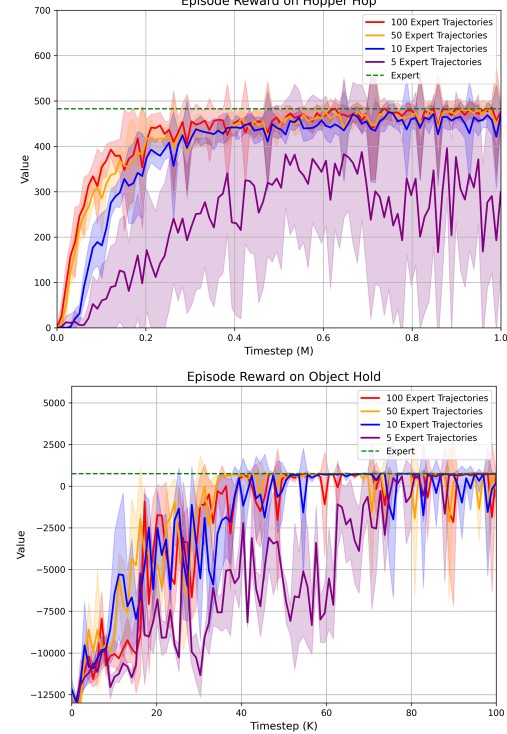

*Figure 5.* **Ablation on Expert Trajectory Numbers.** Performance of IQ-MPC with varying numbers of expert trajectories. Stable expert-level performance is achieved with only 10 expert demonstrations for Hopper Hop (top) and 5 for Object Hold (bottom).

## Impact Statement

This paper presents work whose goal is to advance the field of Machine Learning. There are many potential societal consequences of our work, none of which we feel must be specifically highlighted here.

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

## A. Hyperparameters and Architectural Details

This section will show the detailed hyperparameters and architectures used in our IQ-MPC model.

### A.1. World Model Architecture

All of the components are built using MLPs with Layernorm (Ba, 2016) and Mish activation functions (Misra, 2019). We leverage Dropout for Q networks. The amount of total learnable parameters for IQ-MPC is 4.3M. We depict the architecture in a Pytorch-like notation:

```
Architecture: IQ-MPC(
  (_encoder): ModuleDict(
    (state): Sequential(
      (0): NormedLinear(in_features=state_dim, out_features=256, bias=True, act=Mish)
      (1): NormedLinear(in_features=256, out_features=512, bias=True, act=SimNorm)
    )
  )
  (_dynamics): Sequential(
    (0): NormedLinear(in_features=512+action_dim, out_features=512, bias=True, act=Mish)
    (1): NormedLinear(in_features=512, out_features=512, bias=True, act=Mish)
    (2): NormedLinear(in_features=512, out_features=512, bias=True, act=SimNorm)
  )
  (_pi): Sequential(
    (0): NormedLinear(in_features=512, out_features=512, bias=True, act=Mish)
    (1): NormedLinear(in_features=512, out_features=512, bias=True, act=Mish)
    (2): Linear(in_features=512, out_features=2*action_dim, bias=True)
  )
  (_Qs): Vectorized ModuleList(
    (0-4): 5 x Sequential(
      (0): NormedLinear(in_features=512+action_dim, out_features=512, bias=True, dropout
        =0.01, act=Mish)
      (1): NormedLinear(in_features=512, out_features=512, bias=True, act=Mish)
      (2): Linear(in_features=512, out_features=1, bias=True)
    )
  )
  (_target_Qs): Vectorized ModuleList(
    (0-4): 5 x Sequential(
      (0): NormedLinear(in_features=512+action_dim, out_features=512, bias=True, dropout
        =0.01, act=Mish)
      (1): NormedLinear(in_features=512, out_features=512, bias=True, act=Mish)
      (2): Linear(in_features=512, out_features=1, bias=True)
    )
  )
)
Learnable parameters: 4,274,259
```

The exact parameters above represent the situation when the state dimension is 91, and the action dimension is 39.

Additionally, we also show the convolutional encoder used in our visual experiments:

```
(_encoder): ModuleDict(
    (rgb): Sequential(
      (0): ShiftAug()
      (1): PixelPreprocess()
      (2): Conv2d(9, 32, kernel_size=(7, 7), stride=(2, 2))
      (3): ReLU(inplace=True)
      (4): Conv2d(32, 32, kernel_size=(5, 5), stride=(2, 2))
      (5): ReLU(inplace=True)
      (6): Conv2d(32, 32, kernel_size=(3, 3), stride=(2, 2))
      (7): ReLU(inplace=True)
      (8): Conv2d(32, 32, kernel_size=(3, 3), stride=(1, 1))
      (9): Flatten(start_dim=1, end_dim=-1)
      (10): SimNorm(dim=8)
    )
)
```

## A.2. Hyperparameter Details

The detailed hyperparameters used in IQ-MPC are as follows:

- The batch size during training is 256.

- We balance each part of the loss by assigning weights. For inverse soft Q loss, we assign 0.1. For consistency loss, we assign 20. For the policy and gradient penalty, we assign 1 as the weight.

- We leverage $\lambda = 0.5$ in a horizon.

- We apply the same heuristic discount calculation as TD-MPC2 (Hansen et al., 2023), using 5 as the denominator, with a maximum discount of 0.995 and a minimum of 0.95.

- We iterate 6 times during MPPI planning.

- We utilize 512 samples as the batch size for planning.

- We select 64 samples via top-k selection during MPPI iteration.

- During planning, 24 of the trajectories are generated by the policy prior $\pi$, while normal distributions generate the rest.

- Planning horizon $H = 3$.

- The temperature coefficient is 0.5.

- We set the learning rate of the model to $3e - 4$.

- The entropy coefficient $\beta = 1e - 4$.

- We found no significant improvement by adding the Wasserstein-1 gradient penalty (Eq.14) in locomotion tasks. Therefore, we only apply gradient penalty to manipulation tasks.

- We use $\alpha = 0.5$ for $\chi^2$ divergence $\phi(x) = x - \frac{1}{4\alpha}x^2$.

- We use soft update coefficient $\tau = 0.01$.

## B. Training Algorithm

For completeness, we show the pseudo-code for IQ-MPC training in Algorithm 2.

---
**Algorithm 2** IQ-MPC (*training*)

---
**Require:** $\theta, \theta^-$: randomly initialized network parameters

$\quad\quad\quad\eta, \tau, \lambda, \mathcal{B}_\pi, \mathcal{B}_E$: learning rate, soft update coefficient, horizon discount coefficient, behavioral buffer, expert buffer

$\quad$**for** training steps **do**

$\quad\quad$// *Collect episode with IQ-MPC from* $\mathbf{s}_0 \sim p_0$:

$\quad\quad$**for** step $t = 0...T$ **do**

$\quad\quad\quad$Compute $\mathbf{a}_t$ with $\pi_\theta(\cdot|h_\theta(\mathbf{s}_t))$ using Algorithm 1 $\quad\quad\quad\quad\quad\quad\quad\quad\quad\quad\quad\quad\quad\quad\triangleleft$ *Planning with IQ-MPC*

$\quad\quad\quad(\mathbf{s}'_t, r_t) \sim \text{env.step}(\mathbf{a}_t)$

$\quad\quad\quad\mathcal{B}_\pi \leftarrow \mathcal{B}_\pi \cup (\mathbf{s}_t, \mathbf{a}_t, \mathbf{s}'_t) \quad\quad\quad\quad\quad\quad\quad\quad\quad\quad\quad\quad\quad\quad\quad\quad\quad\quad\quad\quad\triangleleft$ *Add to behavioral buffer*

$\quad\quad\quad\mathbf{s}_{t+1} \leftarrow \mathbf{s}'_t$

$\quad\quad$**end for**

$\quad\quad$// *Update reward-free world model using collected data in* $\mathcal{B}_\pi$ *and* $\mathcal{B}_E$:

$\quad\quad$**for** num updates per step **do**

$\quad\quad\quad(\mathbf{s}_t, \mathbf{a}_t, r_t, \mathbf{s}'_t)_{0:H} \sim \mathcal{B}_\pi \cup \mathcal{B}_E \quad\quad\quad\quad\quad\quad\quad\quad\quad\quad\quad\quad\quad\triangleleft$ *Combine behavioral and expert batch*

$\quad\quad\quad\mathbf{z}_0 = h_\theta(\mathbf{s}_0) \quad\quad\quad\quad\quad\quad\quad\quad\quad\quad\quad\quad\quad\quad\quad\quad\quad\quad\quad\quad\quad\quad\triangleleft$ *Encode first observation*

$\quad\quad\quad$// *Unroll for horizon* $H$

$\quad\quad\quad$**for** $t = 0...H$ **do**

$\quad\quad\quad\quad\mathbf{z}_{t+1} = d_\theta(\mathbf{z}_t, \mathbf{a}_t)$

$\quad\quad\quad\quad\hat{q}_t = Q(\mathbf{z}_t, \mathbf{a}_t)$

$\quad\quad\quad$**end for**

$\quad\quad\quad$Compute critic and consistency loss $\mathcal{L}(\mathbf{z}_{0:H}, \hat{q}_{0:H}, h(\mathbf{s}'_{0:H}), \lambda) \quad\quad\quad\quad\quad\quad\quad\quad\quad\triangleleft$ *Equation 11*

$\quad\quad\quad$Compute policy prior loss $\mathcal{L}_\pi(\mathbf{z}_{0:H}, \lambda) \quad\quad\quad\quad\quad\quad\quad\quad\quad\quad\quad\quad\quad\quad\triangleleft$ *Equation 13*

$\quad\quad\quad$**if** use gradient penalty **then**

$\quad\quad\quad\quad$Compute gradient penalty $\mathcal{L}_{pen}(\mathbf{z}_{0:H}, \mathbf{a}_{0:H}, \lambda) \quad\quad\quad\quad\quad\quad\quad\quad\quad\quad\triangleleft$ *Equation 14*

$\quad\quad\quad$**else**

$\quad\quad\quad\quad\mathcal{L}_{pen} = 0$

$\quad\quad\quad$**end if**

$\quad\quad\quad\theta \leftarrow \theta - \frac{1}{H}\eta\nabla_\theta(\mathcal{L} + \mathcal{L}_\pi + \mathcal{L}_{pen}) \quad\quad\quad\quad\quad\quad\quad\quad\quad\quad\quad\quad\quad\triangleleft$ *Update online network*

$\quad\quad\quad\theta^- \leftarrow (1-\tau)\theta^- + \tau\theta \quad\quad\quad\quad\quad\quad\quad\quad\quad\quad\quad\quad\quad\quad\quad\quad\quad\triangleleft$ *Update target network*

$\quad\quad$**end for**

$\quad$**end for**

---

# C. Task Visualizations

We visualize each task using the random initialization state of an episode. Regarding the locomotion tasks in DMControl, we show them in Figure 6. Figure 7 shows the visualizations of manipulation tasks with dexterous hands in MyoSuite.

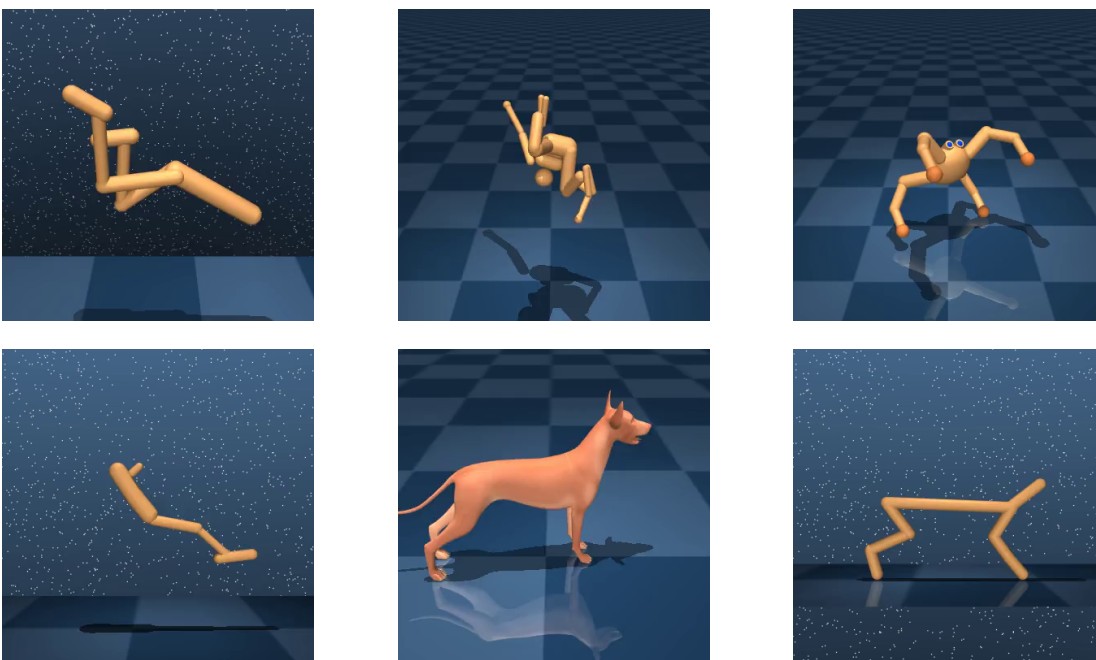

*Figure 6.* **Locomotion Visualizations** The visualizations for DMControl environments, including Hopper, Cheetah, Walker, Quadruped, Humanoid, and Dog.

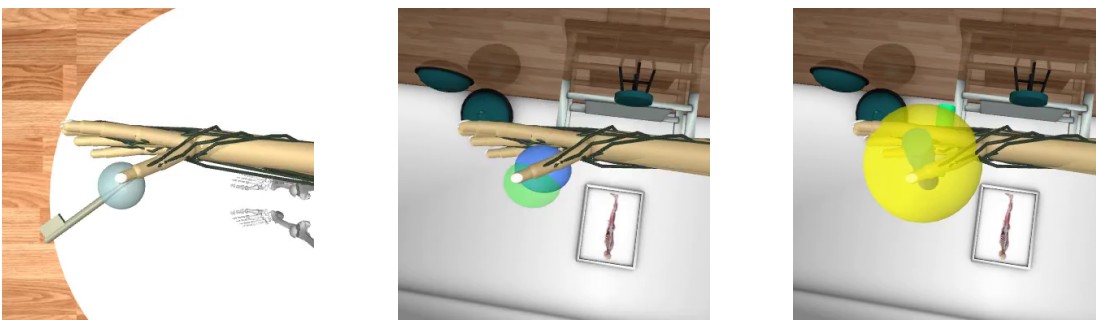

*Figure 7.* **Manipulation Visualizations with Dexterous Hands** The visualizations for MyoSuite tasks, including Key Turn, Object Hold, and Pen Twirl.

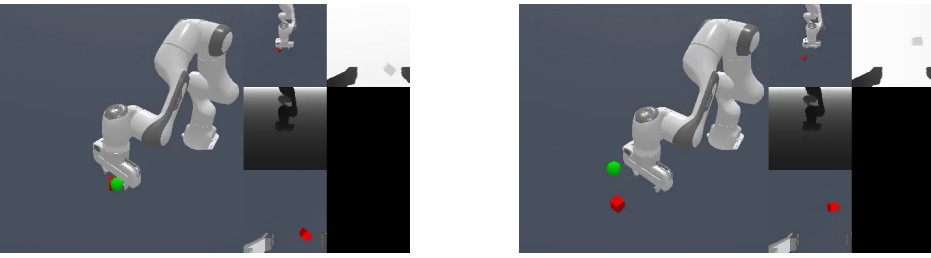

*Figure 8.* **Manipulation Visualizations with Robot Arms** The visualizations for ManiSkill2 tasks, including Pick Cube and Lift Cube.

# D. Environment and Task Details

## D.1. Locomotion Environments

We experiment on 6 locomotion environments in DMControl. The details of the corresponding environments are shown in Table 2. Regarding the visual inverse RL tasks, we take RGB image observations with the shape of $64 \times 64 \times 9$ for inputs. Each observation consists of 3 RGB frames.

| Environment | Observation Dimension | Action Dimension | High-dimensional? |
|---|---|---|---|
| Hopper | 15 | 4 | No |
| Cheetah | 17 | 6 | No |
| Quadruped | 78 | 12 | No |
| Walker | 24 | 6 | No |
| Humanoid | 67 | 24 | Yes |
| Dog | 223 | 38 | Yes |

*Table 2.* **Environment Details for State-based Experiments in DMControl.** We show the environment details for experiments on DMControl with state-based observations. High-dimensional tasks have higher hard levels compared to normal tasks for imitation learning.

## D.2. Manipulation Environment

We experiment on 5 manipulation tasks in ManiSkill2 and MyoSuite. Among these tasks, 2 of them are in ManiSkill2, for which we describe the task details in Table 3, and 3 of them are in MyoSuite, for which we describe the task details in Table 4.

| Task | Observation Dimension | Action Dimension |
|---|---|---|
| Lift Cube | 42 | 4 |
| Pick Cube | 51 | 4 |

*Table 3.* **Task Details for Experiments in ManiSkill2.** We show the environment details for experiments on ManiSkill2. The ManiSkill2 benchmark is built for large-scale robot learning and features extensive randomization and diverse task variations.

| Task | Observation Dimension | Action Dimension |
|---|---|---|
| Object Hold | 91 | 39 |
| Pen Twirl | 83 | 39 |
| Key Turn | 93 | 39 |

*Table 4.* **Task Details for Experiments in MyoSuite.** We show the environment details for experiments on MyoSuite. The MyoSuite benchmark is designed for physiologically accurate, high-dimensional musculoskeletal motor control, featuring highly complex object manipulation using a dexterous hand.

# E. Additional Experiments

## E.1. Additional High-dimensional Locomotion Experiments

To show the robustness of our model in high-dimensional tasks, we conduct locomotion experiments on the Dog environment with different tasks such as standing, trotting, and walking, in addition to the running task in Section 5.1.1. The dog environment is a relatively complex environment due to its high-dimensional observation and action spaces. We leverage 500 expert trajectories sampled from trained TD-MPC2 for each experiment. We show the results in Figure 9.

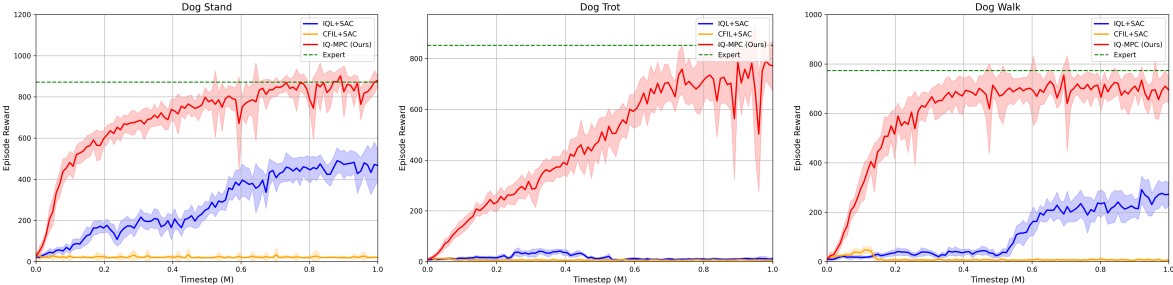

*Figure 9.* **Additional High-dimensional Locomotion Experiments** Our IQ-MPC shows stable and expert-level performance on different tasks in the Dog environment, which demonstrates our model's capability in handling high-dimensional tasks. In the plots, the blue lines and orange lines represent the results from IQL+SAC (Garg et al., 2021) and CFIL+SAC (Freund et al., 2023), respectively, while the red lines represent the results from our IQ-MPC.

## E.2. Additional Manipulation Experiments

We also evaluate our method on simpler manipulation tasks in ManiSkill2 (Gu et al., 2023). We show stable and comparable results in the pick cube task and lift cube task. IQL+SAC (Garg et al., 2021) also performs relatively well in these simple settings. Figure 10 shows the episode rewards results in ManiSkill2 tasks, and Table 5 demonstrates the success rate of each method.

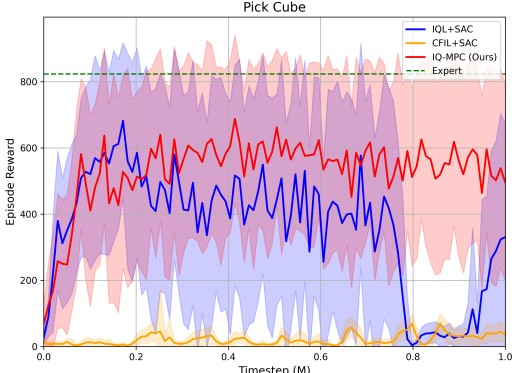 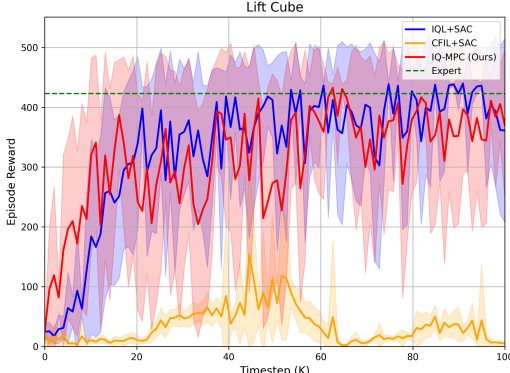

*Figure 10.* **Manipulation Results in ManiSkill2** Our IQ-MPC shows stable and comparable results in ManiSkill2 manipulation experiments. In the plots, the color settings are the same as those in Figure 9.

| Method | IQL+SAC | CFIL+SAC | IQ-MPC(Ours) |
|---|---|---|---|
| Pick Cube | 0.61±0.13 | 0.00±0.00 | **0.79±0.05** |
| Lift Cube | 0.85 ± 0.04 | 0.01±0.01 | **0.89±0.02** |

*Table 5.* **Manipulation Success Rate Results in ManiSkill2** We evaluate the success rate of IQ-MPC on pick and lift tasks in the ManiSkill2 environment. We show outperforming empirical results compared to IQL+SAC, and CFIL+SAC. We show the results by averaging over 100 trajectories and evaluating over 3 random seeds.

### E.3. Additional Ablation Studies

In this section, we demonstrate the results of ablating over objective formulation, gradient penalty, and hyperparameter $\alpha$.

**Objective Formulation**   We observe performance improvement using the reformulated objective Eq.12 as we mentioned in the Model Learning part of Section 4.1. In details, we changed the value temporal difference term $\mathbb{E}_{(\mathbf{s}_t,\mathbf{a}_t,\mathbf{s}'_t)\sim\mathcal{B}_\pi}[V^\pi(\mathbf{z}_t) - \gamma V^\pi(\mathbf{z}'_t)]$ into a form only containing value from initial distribution $\mathbb{E}_{\mathbf{s}_0\sim\mathcal{B}_E}[(1-\gamma)V^\pi(\mathbf{z}_0)]$. This technique is also mentioned in the original IQ-Learn paper (Garg et al., 2021). We have given the theoretical proof for mathematical equivalence in Lemma H.3. In this section, we provide the empirical analysis regarding the effectiveness of this technique in the context of our IQ-MPC model. We observe stabler Q estimation leveraging this technique. Moreover, in this case, the difference in Q estimation between the expert batch and the behavioral batch can converge more easily, especially for high-dimensional cases like the Humanoid Walk and Dog Run task. The better convergence of Q estimation difference shows that the Q function faces difficulty in distinguishing between expert and behavioral demonstrations, which implies that the policy prior behaves similarly as expert demonstrations. The stable Q estimation results in a better learning behavior for the latent dynamics model, which is observed by measuring the prediction consistency loss (The first term in Eq.11) during training. The results in Humanoid Walk task are shown in Figure 11.

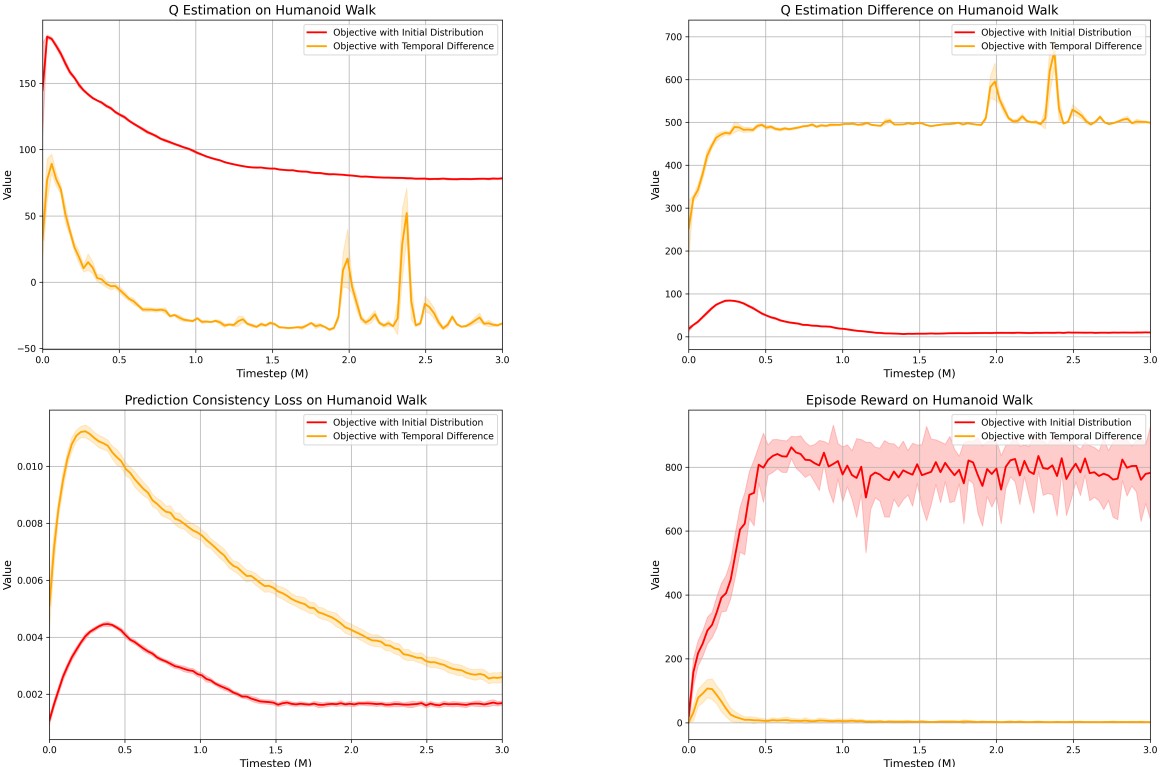

*Figure 11.* **Ablation on Objective Formulation** We show that the Q estimation and training dynamics are stabler by utilizing objective with initial distribution compared to leveraging objective with temporal difference. Moreover, we obtain stable expert-level performance leveraging the objective with the initial distribution. We depict the stability by showing plots regarding Q estimation and prediction consistency. The red lines are the stabler results using the objective with initial distribution while orange lines are the results with temporal difference objective. The ablation experiments are conducted on the Humanoid Walk task.

**Gradient Penalty**    We ablate over the Wasserstein-1 metric gradient penalty in Eq.14 with our experiments. This training technique balances the discriminative power of the Q network to ensure stable policy learning. We show improvement in training stability on the Pick Cube task in the ManiSkill2 environment. By leveraging gradient penalty, we observe stable convergence regarding the difference in Q estimation between expert and behavioral batch. This behavior results in stabler policy learning, especially in tasks with low dimensional state or action space, where expert and behavioral demonstrations can be easily distinguished. The ablation results are shown in Figure 12 and Table 6.

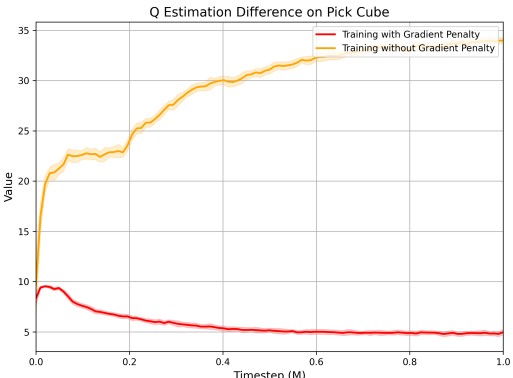 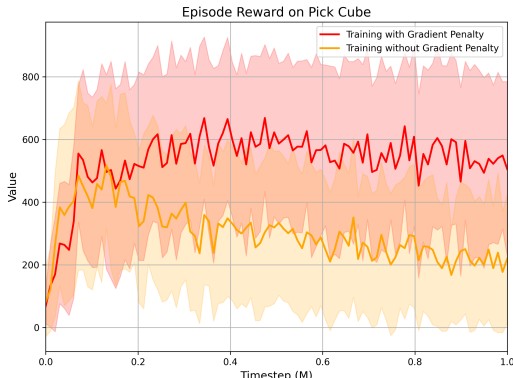

*Figure 12.* **Ablation on Gradient Penalty.** We show the improvement by adopting the Wasserstein-1 gradient penalty by demonstrating the effect over the convergence of Q-difference, which is the difference between Q estimation on expert and behavioral demonstrations. The converging Q-difference implies stable policy learning and reasonable discriminative power of the Q network. We also demonstrate the effectiveness of the gradient penalty by episode reward during training. The red lines represent results with gradient penalty while orange lines represent results without it.

| Gradient Penalty? | Yes | No |
|---|---|---|
| Success Rate | **0.79±0.05** | 0.51±0.11 |

*Table 6.* **Ablation on Gradient Penalty with Success Rate** We evaluate the success rate of IQ-MPC with and without gradient penalty on ManiSkill2 Pick Cube task. We show the results by averaging over 100 trajectories and evaluating over 3 random seeds.

**Hyperparameter Selection**    We perform an ablation study on the selection of the hyperparameter $\alpha$ in Eq.12. The hyperparameter $\alpha$ controls the strength of the $\chi^2$ regularization applied to the inverse soft-Q objective. Intuitively, the last term in Eq.12 serves as a penalty on the magnitude of the estimated reward. Therefore, smaller values of $\alpha$ result in a larger penalty on the estimated reward magnitude, which helps enforce training stability and prevents Q estimation from exploding. In contrast, larger values of $\alpha$ encourage more aggressive estimation of the reward and Q value, increasing the chances of training instability. We experiment with the effect of this hyperparameter in the Humanoid Walk task and conclude that $\alpha = 0.5$ is the optimal choice. We present our results in Figure 13.

### E.4. Experiments on Noisy Environment Dynamics

In this section, we evaluate the robustness of our IQ-MPC model under noisy and stochastic environment dynamics. HyPE (Ren et al., 2024) has demonstrated relatively robust performance when subjected to minor noise perturbations in environment transitions. Although our model is primarily designed for fully deterministic settings, we observe that it exhibits a degree of robustness when handling stochastic environment dynamics.

For our experiments, we adopt the same environment settings as HyPE (Ren et al., 2024), introducing a trembling noise probability, $p_{tremble}$, into the environment transitions. Specifically, we assess the impact of $p_{tremble}$ on our IQ-MPC model in the Walker Run task. The results indicate that our model maintains certain robustness even in the presence of transition noise. We present the results the Figure 14.

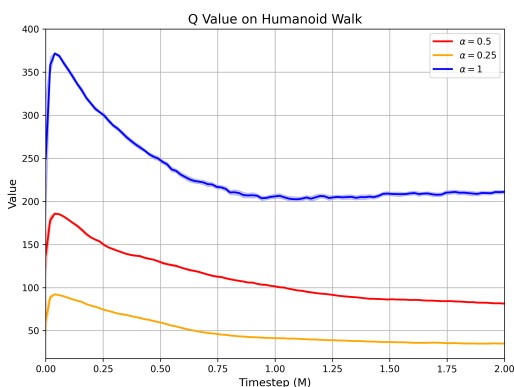
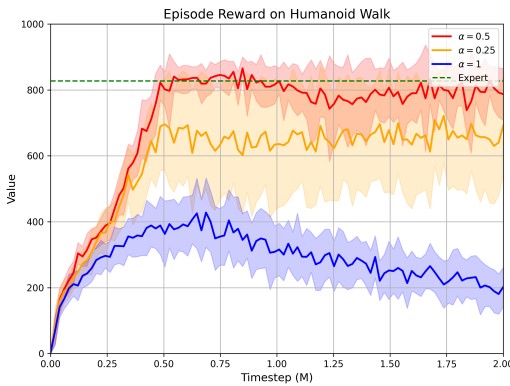

*Figure 13.* **Ablation on Hyperparameter Selection.** In our ablation study, we found that larger values of $\alpha$ may lead to higher Q estimations, resulting in suboptimal and unstable training behavior. Conversely, smaller values of $\alpha$ lead to lower Q estimations, which correspond to stronger regularization and may also cause suboptimal performance.

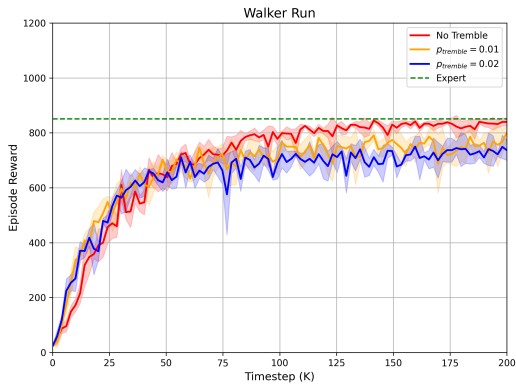
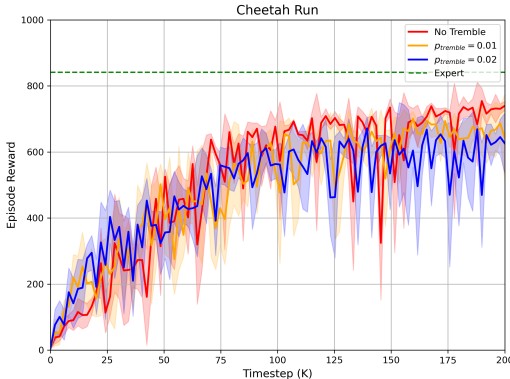

*Figure 14.* **Experiments on Noisy Environment Dynamics** We evaluate our model's performance on the Walker Run and Cheetah Run task under different values of $p_{tremble}$, where a larger $p_{tremble}$ indicates greater stochasticity in the environment dynamics. Specifically, we experiment with $p_{tremble} = 0.01$ and $p_{tremble} = 0.02$, observing only slight performance degradation. This suggests that our model exhibits a degree of robustness to noisy environment dynamics.

### E.5. Additional Experiments with Few Expert Demonstrations on Visual Tasks

In this section, we demonstrate that our approach can also learn visual tasks using only 10 expert demonstrations. Results with 10 demonstrations are shown in Figure 15. While the convergence is slower, successful learning is still achievable under this low-data regime.

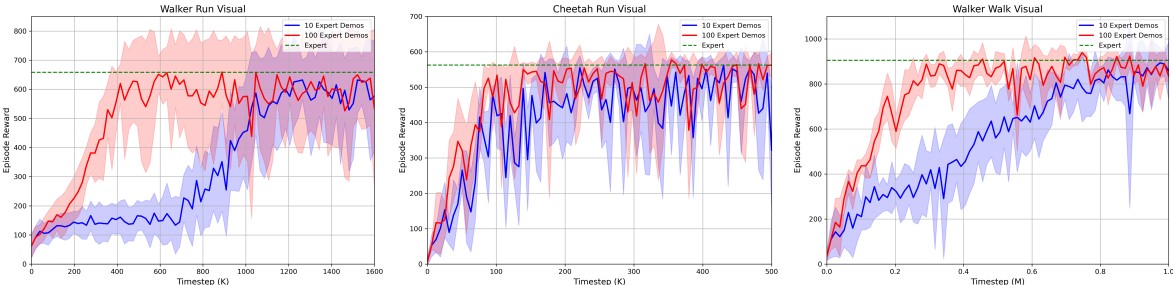

*Figure 15.* **Additional Experiments with Few Expert Demonstrations on Visual Tasks** We show that our IQ-MPC method can successfully learn from only 10 expert demonstrations in DMControl visual tasks. We present results on the Walker Run, Cheetah Run, and Walker Walk tasks, and include performance curves with 100 expert demonstrations as a reference.

## F. Training Time Evaluation

We evaluate the computational overhead of our approach in comparison to model-free baselines (IQL+SAC (Garg et al., 2021), CFIL+SAC (Freund et al., 2023), HyPE (Ren et al., 2024)) and the model-based baseline (HyPER (Ren et al., 2024)). Additionally, we assess the training time of IQ-MPC with model predictive control enabled and when interacting solely with the policy prior. The experiments are conducted on the Humanoid Walk task, with training time reported in seconds. All baselines are trained using a single RTX 2080 Ti GPU. The results are shown in Figure 16.

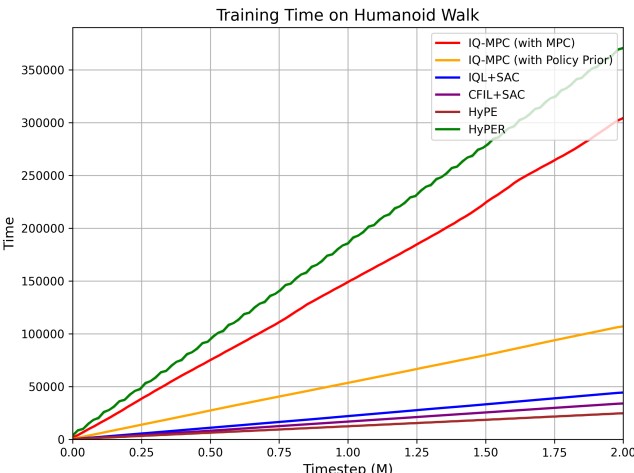

*Figure 16.* **Computational Overhead** We evaluate the computational cost during training of our model on the Humanoid task. Leveraging a policy prior for direct interaction, instead of relying on MPC, accelerates the training process but may introduce greater instability. Our model requires less computational time compared to HyPER, although its training remains slower than model-free baselines.

## G. Reward Correlation Analysis

We evaluate the ability to recover rewards using a trained IQ-MPC model, which demonstrates our model's capability of handling inverse RL tasks. We observe a positive correlation between ground-truth rewards and our recovered rewards. We conduct this experiment on the DMControl Cheetah Run task and decode rewards via $r(\mathbf{z}, \mathbf{a}) = Q_\theta(\mathbf{z}, \mathbf{a}) - \gamma \mathbb{E}_{\mathbf{z}' \sim d_\theta(\cdot|\mathbf{z},\mathbf{a})} V^\pi(\mathbf{z}')$. We evaluate over 5 trajectories sampled from a trained IQ-MPC. The results are revealed in Figure 17.

We further analyze the correlation between the decoded rewards and ground-truth rewards in the Hopper Hop, Cheetah Run, Quadruped Run, and Walker Run tasks. Specifically, we compute the Pearson correlation between the estimated and ground-truth rewards in these settings, using IQL+SAC (Garg et al., 2021) as the comparison baseline. The results are presented in Table 7.

In Figure 17, we observe that the variance of the estimated rewards is higher when the ground-truth reward is high. One possible explanation for this high variance in the estimated expert rewards is as follows:

There are multiple equivalent reward formulations that result in optimal trajectories, and the maximum entropy objective selects the one with the highest entropy. Our actor-critic architecture, optimized with the maximum entropy inverse RL objective, leads to a more evenly distributed reward structure for expert demonstrations. Consequently, rewards closer to the expert tend to exhibit higher variance, a phenomenon also observed in (Freund et al., 2023).

## H. Additional Theoretical Supports

We first give a proper definition of distributions involving latent state representations:

**Definition H.1.** Define a latent state distribution $\tilde{p}_t^\pi = h_* p_t^\pi$ as a pushforward distribution of original state distribution $p_t^\pi$ for policy $\pi$ given an encoder mapping $h : \mathcal{S} \to \mathcal{Z}$.

**Definition H.2.** Define a latent state-action distribution with policy $\pi$ as $\tilde{\rho}^\pi$ on $\mathcal{Z} \times \mathcal{A}$ from an original state-action distribution $\rho^\pi$ on $\mathcal{S} \times \mathcal{A}$ with an encoder mapping $h : \mathcal{S} \to \mathcal{Z}$.

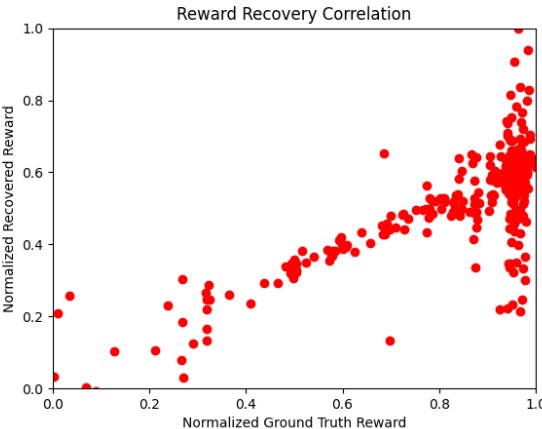

*Figure 17.* **Reward Recovery.** The IQ-MPC model successfully recovers rewards in the inverse RL setting, showing a positive correlation with ground-truth rewards. This experiment is conducted on the Cheetah Run task with state-based observations from DMControl.

| Method | IQL+SAC | IQ-MPC (Ours) |
|---|---|---|
| Hopper Hop | 0.49 | **0.88** |
| Cheetah Run | 0.79 | **0.87** |
| Walker Run | 0.65 | **0.91** |
| Quadruped Run | 0.88 | **0.93** |

*Table 7.* **Pearson Correlations of Reward Recovery** We evaluate the Pearson correlation between the decoded rewards from IQL+SAC and IQ-MPC in the Hopper Hop, Cheetah Run, Quadruped Run, and Walker Run tasks. Our results demonstrate that IQ-MPC achieves a higher correlation with ground-truth rewards when trained on these tasks.

### H.1. Objective Equivalence

In this section, we will provide proof for the reformulation of the second term in Eq.12 for completeness. We borrow the proof from (Garg et al., 2021) and slightly modify it to fit our setting with latent representations instead of actual states. The proof is demonstrated in Lemma H.3. In Eq.12, we use the mean over encoded latent representation batch sampled from the expert buffer $\mathcal{B}_E$ to approximate the mean over initial distribution $\tilde{p}_0$ on latent representation.

**Lemma H.3** (Objective Equivalence). *Given a latent transition model $d(\mathbf{z}'|\mathbf{z}, \mathbf{a})$, a latent state distribution $\tilde{p}_t^\pi$ for time step $t$ and a latent state-action distribution $\tilde{\rho}^\pi$, we have:*

$$\mathbb{E}_{(\mathbf{z},\mathbf{a})\sim\tilde{\rho}_\pi}[V^\pi(\mathbf{z}) - \gamma\mathbb{E}_{\mathbf{z}'\sim d(\cdot|\mathbf{z},\mathbf{a})}V^\pi(\mathbf{z}')] = (1-\gamma)\mathbb{E}_{\mathbf{z}_0\sim\tilde{p}_0}[V^\pi(\mathbf{z}_0)]$$

*Proof.* We decompose the left-hand side into a summation:

$$\mathbb{E}_{(\mathbf{z},\mathbf{a})\sim\tilde{\rho}_\pi}[V^\pi(\mathbf{z}) - \gamma\mathbb{E}_{\mathbf{z}'\sim d(\cdot|\mathbf{z},\mathbf{a})}V^\pi(\mathbf{z}')]$$

$$= (1-\gamma)\sum_{t=0}^{\infty}\gamma^t\mathbb{E}_{\mathbf{z}\sim\tilde{p}_t^\pi,\mathbf{a}\sim\pi(\mathbf{z})}[V^\pi(\mathbf{z}) - \gamma\mathbb{E}_{\mathbf{z}'\sim d(\cdot|\mathbf{z},\mathbf{a})}V^\pi(\mathbf{z}')]$$

$$= (1-\gamma)\sum_{t=0}^{\infty}\gamma^t\mathbb{E}_{\mathbf{z}\sim\tilde{p}_t^\pi}[V^\pi(\mathbf{z})] - (1-\gamma)\sum_{t=0}^{\infty}\gamma^{t+1}\mathbb{E}_{\mathbf{z}\sim\tilde{p}_{t+1}^\pi}[V^\pi(\mathbf{z})]$$

$$= (1-\gamma)\mathbb{E}_{\mathbf{z}_0\sim\tilde{p}_0}[V^\pi(\mathbf{z}_0)]$$

$\square$

## H.2. Policy Update Guarantee

We prove that policy update objective Eq.13 can search for the saddle point in optimization, which increases $\mathcal{L}_{iq}(\pi, Q)$ with $Q$ fixed, following (Garg et al., 2021). For simplicity, we prove it with horizon $H = 1$, and it's generalizable to objective with discounted finite horizon.

**Theorem H.4** (Policy Update). *Updating the policy prior via maximum entropy objective increases $\mathcal{L}_{iq}(\pi, Q)$ with $Q$ fixed. We assume entropy coefficient $\beta = 1$.*

*Proof.* For a fixed $Q$:

$$V^{\pi}(\mathbf{z}) = \mathbb{E}_{a \sim \pi(\cdot | \mathbf{z})}[Q(\mathbf{z}, \mathbf{a}) - \log(\pi(\mathbf{a} | \mathbf{z}))]$$
$$= -D_{KL}\Big(\pi(\cdot | \mathbf{z}) \Big\| \frac{\exp Q(\mathbf{z}, \cdot)}{\sum_{\mathbf{a}} \exp(Q(\mathbf{z}, \mathbf{a}))}\Big) + \log\Big(\sum_{\mathbf{a}} \exp(Q(\mathbf{z}, \mathbf{a}))\Big)$$

Policy update with maximum entropy objective is optimizing:

$$\pi^* = \operatorname{argmin}_{\pi} D_{KL}\Big(\pi(\cdot | \mathbf{z}) \Big\| \frac{\exp Q(\mathbf{z}, \cdot)}{\sum_{\mathbf{a}} \exp(Q(\mathbf{z}, \mathbf{a}))}\Big)$$

Assume that we have an updated policy $\pi'$ via gradient descent with learning rate $\xi$:

$$\pi' = \pi - \xi \, \nabla_{\pi} D_{KL}\Big(\pi(\cdot | \mathbf{z}) \Big\| \frac{\exp Q(\mathbf{z}, \cdot)}{\sum_{\mathbf{a}} \exp(Q(\mathbf{z}, \mathbf{a}))}\Big)$$

We can obtain $V^{\pi}(\mathbf{z}) < V^{\pi'}(\mathbf{z})$. In regions where $\phi(x)$ is monotonically non-decreasing and $Q$ is fixed, we can have $\mathcal{L}_{iq}(\pi', Q) > \mathcal{L}_{iq}(\pi, Q)$. $\qquad\square$

## H.3. Analysis on the Consistency Loss

We provide a more detailed analysis regarding the relationship between minimizing the consistency loss in Eq.11 and minimizing T2 in the bound provided by Lemma 4.1.Specifically, our consistency loss directly minimizes the upper bound of T2 under following assumptions:

**Assumption H.5.** The latent dynamics $d : \mathcal{Z} \times \mathcal{A} \to \Delta_{\mathcal{Z}}$ is approximately a Gaussian distribution on latent space $\mathcal{Z}$ with $\mathcal{N}(\mu_d, \sigma_d^2)$.

**Assumption H.6.** The standard deviation of our learned latent dynamics $\hat{\sigma}_d$ is close to the actual standard deviation $\sigma_d$.

Considering T2 in Lemma 4.1 and neglecting the constant coefficient, according to Pinsker Inequality, we have:

$$\mathbb{E}_{\rho_{\mathcal{M}}^{\pi}}\Big[D_{TV}(d(\mathbf{z}' | \mathbf{z}, \mathbf{a}), \hat{d}(\mathbf{z}' | \mathbf{z}, \mathbf{a}))\Big] \le \mathbb{E}_{\rho_{\mathcal{M}}^{\pi}} \sqrt{\frac{1}{2} D_{KL}(d(\mathbf{z}' | \mathbf{z}, \mathbf{a}), \hat{d}(\mathbf{z}' | \mathbf{z}, \mathbf{a}))}$$

With Assumption H.5 and H.6, we can represent the KL divergence by mean and standard deviation of actual and learned latent dynamics:

$$D_{KL}(d(\mathbf{z}' | \mathbf{z}, \mathbf{a}), \hat{d}(\mathbf{z}' | \mathbf{z}, \mathbf{a})) = \log \frac{\hat{\sigma}_d}{\sigma_d} + \frac{\sigma_d^2 + (\mu_d - \hat{\mu}_d)^2}{2 \hat{\sigma}_d^2} - \frac{1}{2} \approx \frac{(\mu_d - \hat{\mu}_d)^2}{2 \sigma_d^2}$$

Given a predicted latent state $\hat{z}'$ from the learned dynamics $\hat{d}$ and an actual latent state $z' = h(s')$ encoded from the next state observation with unknown dynamics $d$, minimizing the L2 loss approximately minimizes the distance between the means of the learned and actual latent dynamics distributions. This, in turn, minimizes the right-hand side of the Pinsker inequality under our assumptions. Consequently, our consistency loss minimizes the statistical distance between the dynamics.

# I. Limitations and Future Work

One limitation of our method is its sensitivity to the size of the observation and action spaces. Empirically, we find that in scenarios with low-dimensional observations or actions, the discriminator can become overly powerful, potentially leading to training instability, as discussed in Section 4.1. While we address this issue by incorporating a Wasserstein-1 gradient penalty, it may not be sufficient in all settings. Future work could explore more robust stabilization techniques tailored to tasks with small observation or action space dimensions. Additionally, applying our method to real-world robotic tasks would be a valuable direction to assess its practical effectiveness and generalization capabilities.

