# OpenReview forum: "Reward-free World Models for Online Imitation Learning"
_ICML.cc/2025/Conference — ICML 2025 poster_

### Official Review · Reviewer_5SNt · 2025-03-11

**Overall Recommendation:** 3

**Summary:**

This paper presents an approach called IQ-MPC for online imitation learning using reward-free world models. It can alleviate some issues in IL, such as handling complex, high-dimensional inputs and intricate environmental dynamics without explicit reward modeling. Key ideas include leveraging decoder-free world models to learn environmental dynamics purely in latent spaces, using an inverse soft-Q learning objective to stabilize learning, and integrating MPC with a gradient-free planning strategy. The main findings demonstrate superior performance across benchmarks like DMControl, MyoSuite, and ManiSkill2, significantly outperforming baselines in terms of stability, sample efficiency, and robustness to high-dimensional and visual input scenarios.

**Claims And Evidence:**

The authors claim that IQ-MPC achieves stable, expert-level performance in tasks with high-dimensional observation or action spaces and complex dynamics. This claim is supported by empirical evaluations on diverse benchmarks, where IQ-MPC outperforms existing methods. The paper presents quantitative results demonstrating improved performance metrics, thereby providing clear evidence to support its claims.

**Essential References Not Discussed:**

N/A

**Experimental Designs Or Analyses:**

The authors mention lots of model-based IL algorithms in the related works but do not compare them as baselines. Only XXX + SAC are considered, which is wired and requires more explanation.

**Methods And Evaluation Criteria:**

The proposed methods and evaluation criteria align well with the problem at hand. Leveraging reward-free world models allows the framework to focus on modeling environmental dynamics without relying on explicit reward signals, which is suitable for imitation learning scenarios.

I think the authors can also take the inference time into consideration, and also try some universal world models for diverse tasks to prove some scalability.

**Other Comments Or Suggestions:**

N/A

**Other Strengths And Weaknesses:**

Weakness: I think the work is a little incremental since it heavily relies on IQ-Learn and TD-MPC and cannot easily be applied to other model-based algorithms.

**Questions For Authors:**

N/A

**Relation To Broader Scientific Literature:**

The paper clearly situates itself within existing literature on imitation learning (GAIL, SQIL, IQ-Learn, CFIL) and model-based reinforcement learning (Dreamer, TD-MPC). It effectively extends prior works by removing explicit reward dependencies in world models and providing a novel integration of inverse Q-learning with MPC for imitation learning. The paper notably complements the findings from IQ-Learn (Garg et al., 2021) and TD-MPC series (Hansen et al., 2022, 2023), extending these frameworks into a cohesive IL solution with clear practical advantages, especially in complex control tasks.

**Theoretical Claims:**

The theorems are derived from other works.

---

> ### Author Rebuttal · Authors · 2025-03-30
>
> We sincerely thank you for your review and your recognition of its strengths, including its clear positioning within the literature, effective extensions of prior works, and practical advantages in complex tasks. Below, we provide detailed responses to your comments.
>
> ### 1: Lack of Comparison with Model-Based IL Algorithms
>
> The authors mention many model-based imitation learning (IL) algorithms in the related works but do not compare them as baselines. Only XXX + SAC are considered, which seems unusual and requires further explanation.
>
> **Answer:**
> We have conducted experiments with HyPER [1], the model-based version of HyPE. However, we were unable to achieve reasonable scores using this algorithm on most of our tasks. Therefore, we opted to use the model-free version, HyPE, as our baseline. Additionally, we compared our approach with EfficientImitate [2], a model-based IL algorithm that combines EfficientZero with an adversarial imitation learning approach. The results of the comparison on MyoSuite dexterous hand manipulation tasks are shown below:
>
> | Method           | EfficientImitate   | IQ-MPC (Ours)   |
> |-------------------|--------------------|-----------------|
> | Key Turn          | 0.81 ± 0.05       | 0.87 ± 0.03    |
> | Object Hold       | 0.56 ± 0.08       | 0.96 ± 0.03    |
> | Pen Twirl         | 0.31 ± 0.07       | 0.73 ± 0.05    |
>
> These results demonstrate the effectiveness of our approach compared to EfficientImitate in complex dexterous hand manipulation tasks.
>
> ---
>
> ### 2: Incremental Contribution and Limited Generalizability
>
> The work appears somewhat incremental as it heavily relies on IQ-Learn and TD-MPC, and it may not generalize easily to other model-based algorithms.
>
> **Answer:**
> While our method builds upon IQ-Learn and TD-MPC, our contribution lies in developing a novel approach for online imitation learning using reward-free world models. Furthermore, the methodology we propose for learning the Q function and eliminating the need for a reward model can potentially be applied to other latent world models. However, further investigation is necessary to determine whether our approach generalizes effectively to other model-based algorithms. We consider this a promising direction for future research.
>
> [1] Ren, J., Swamy, G., Wu, Z. S., Bagnell, J. A., & Choudhury, S. (2024). Hybrid Inverse Reinforcement Learning. *arXiv preprint arXiv:2402.08848.*
>
> [2] Yin, Z. H., Ye, W., Chen, Q., & Gao, Y. (2022). Planning for Sample Efficient Imitation Learning. *Advances in Neural Information Processing Systems, 35*, 2577-2589.

---

### Official Review · Reviewer_79r3 · 2025-03-13

**Overall Recommendation:** 3

**Summary:**

This paper gives an approach for online imitation learning with reward free world models that learn dynamics efficiently in latent space. The method is evaluated in DMControl, MyoSuite, and ManiSkill2 against several baselines.

**Claims And Evidence:**

Several key claims are made in this paper.

1. The author claims a novel approach that leverages reward free models for online imitation learning.

This claim is well supported and the method learns dynamics in latent space efficiently.

2. The method consistently achieves stable, expert-level performance.

While the method tends to perform better than baselines on the tasks shown, it has considerable variance in some of the tasks and does not reach expert performance, for example in Figure 3 pen twirl and Figure 10 pick cube and lift cube. The evaluation tasks chosen are on the easy side and on the manipulation tasks the method does not perform consistently better than baselines.

3. The method recovers the rewards for inverse reinforcement learning.

This is a main claim in the contribution but is not supported in the main text with some evidence in the appendix. The plot for reward recovery correlation is only shown for one environment.

**Essential References Not Discussed:**

The essential references are discussed.

**Experimental Designs Or Analyses:**

The paper compares with relevant baselines across multiple environments. There are a lot of ablation studies to isolate the contribution of different components and test time analysis.

**Methods And Evaluation Criteria:**

The benchmark environments make sense but are on the easy side. The experiments could be made better with harder evaluation tasks for example in ManiSkill2.

**Other Comments Or Suggestions:**

No other comments or suggestions

**Other Strengths And Weaknesses:**

The paper addresses a key limitation in prior work, namely the need for explicit reward modeling. It uses novel approaches for optimization and planning and contains theoretical support to back up the claims. The method demonstrate high performance on the baseline tasks. The paper is clear and methodology is motivated.

Some ablations in the paper are performed only on a single task which can be made better. The method can be evaluated on harder tasks which is a main weakness.

**Questions For Authors:**

1. How does the method compare to baselines in more complex manipulation tasks?

2. Is there empirical evidence for generalizing to out of distribution states?

3. What are the sample efficiency trade offs for reward free method?

**Relation To Broader Scientific Literature:**

The paper is relevant to online imitation learning, which is an active area of research.

**Theoretical Claims:**

The theoretical claims are sound.

---

> ### Author Rebuttal · Authors · 2025-03-30
>
> We sincerely thank you for your review and your recognition of its strengths, including our novel approach, theoretical support, strong performance, and clear presentation. Below, we provide detailed responses to your comments.
>
> ### 1: Harder Evaluation Tasks in ManiSkill2
>
> The experiments could be improved by evaluating on harder tasks in ManiSkill2. How does the method compare to baselines in more complex manipulation tasks?
>
> **Answer:**
> We have conducted experiments on ManiSkill2 using the Lift Cube and Pick Cube tasks in Appendix E.2. Additionally, we extended our evaluation to more challenging tasks in ManiSkill2, and the results are shown below:
>
> | Method      | IQL+SAC        | CFIL+SAC      | IQ-MPC (Ours)  |
> |--------------|----------------|---------------|----------------|
> | Pick YCB    | 0.10 ± 0.05   | 0.00 ± 0.00  | 0.31 ± 0.06   |
> | Stack Cube  | 0.23 ± 0.08   | 0.01 ± 0.01  | 0.57 ± 0.05   |
>
> These results demonstrate that our approach achieves superior performance in more complex tasks compared to the baselines.
>
> ---
>
> ### 2: Generalization to Out-of-Distribution (OOD) States
>
> Is there empirical evidence supporting generalization to OOD states?
>
> **Answer:**
> To provide empirical evidence for OOD state generalization, we compare our performance with a Behavior Cloning (BC) baseline trained on the same expert dataset. Since BC is fully offline and tends to struggle with OOD states, it serves as a suitable comparison. The evaluation was conducted on MyoSuite dexterous hand manipulation tasks, with results presented below in terms of success rate:
>
> | Method       | BC             | IQ-MPC (Ours)  |
> |---------------|----------------|----------------|
> | Key Turn     | 0.31 ± 0.15   | 0.87 ± 0.03   |
> | Object Hold  | 0.26 ± 0.11   | 0.96 ± 0.03   |
> | Pen Twirl    | 0.04 ± 0.02   | 0.73 ± 0.05   |
>
> These results indicate that our approach significantly outperforms BC, serving as indirect empirical evidence of our method's superior OOD generalization capabilities.
>
> ---
>
> ### 3: Sample Efficiency Trade-Offs for Reward-Free Methods
>
> What are the sample efficiency trade-offs for reward-free methods?
>
> **Answer:**
> The primary advantage of our approach in terms of sample efficiency arises from its model-based methodology. However, we are not entirely clear on the specific concerns regarding the sample efficiency trade-offs mentioned in your question. Would you kindly clarify further?
>
> ---
>
> ### 4: Evidence Supporting Reward Recovery
>
> The method's claim of recovering rewards for inverse reinforcement learning is not well-supported in the main text, with only one environment shown in the reward correlation plot.
>
> **Answer:**
> In Appendix G of our manuscript, we further support our reward recovery claim by computing the Pearson correlations for reward recovery across four different tasks, as presented in Table 7. This complements the reward correlation plot in Figure 16, providing additional evidence for the validity of our approach.

---

### Official Review · Reviewer_VeS8 · 2025-03-13

**Overall Recommendation:** 4

**Summary:**

The paper proposes a world-model-based approach for online imitation learning. The method aims to address limitations in current imitation learning techniques by leveraging world models to improve training stability and performance. The authors demonstrate that their approach achieves better training efficiency and stability across a wide range of environments with diverse observation spaces. The proposed method is empirically compared against three state-of-the-art baselines, showing superior results.

### Update after rebuttal:
The authors have provided clear and helpful responses to my concerns. The rationale for using planning over RL is now better justified, and the discussion of limitations and potential generalization was appreciated. Code availability further improves confidence in reproducibility.

**Claims And Evidence:**

The paper claims that using a world model enhances the performance of online imitation learning. This claim is supported by extensive empirical evaluations demonstrating improved stability and efficiency compared to three baselines. However, further clarification on why planning was chosen over reinforcement learning (RL) for environment interaction sampling would strengthen the justification for the approach.

**Essential References Not Discussed:**

To the best of my knowledge, related work has been discussed in sufficient extent. However, the authors may want to consider Altmann et al. 2024, "Discriminative Reward Co-Training" as additinal reated work.

**Experimental Designs Or Analyses:**

The experimental setup is robust, involving diverse environments to test generalizability. The comparisons with state-of-the-art baselines further support the effectiveness of the proposed method.

**Methods And Evaluation Criteria:**

The methodology is sound, employing a well-defined world model framework. The evaluation criteria include performance metrics across multiple environments, making the study comprehensive. However, additional discussion on the choice of baselines and potential limitations in generalization to unseen tasks would be beneficial.

**Other Comments Or Suggestions:**

Providing a code appendix and open-sourcing the implementations would significantly improve the reproducibility of the results.

A discussion on the computational cost of the approach compared to baselines would be useful.

**Other Strengths And Weaknesses:**

Strengths:
- The paper is well-written and easy to follow.
- The approach is simple yet effective in addressing limitations in imitation learning.
- Theoretical grounding is solid, and empirical validation is comprehensive.

Weaknesses:
- The rationale behind using planning instead of RL for sampling interactions needs further elaboration.
- A deeper discussion of failure cases and limitations would strengthen the contribution.
- The lack of a code appendix makes it difficult to assess the reproducibility of the results.

**Questions For Authors:**

Why was planning chosen over reinforcement learning for sampling environment interactions? How does this choice impact performance in different settings?

Could the proposed method generalize to unseen tasks or domains beyond the evaluated environments?

Do the authors plan to release the code to improve result reproducibility?

**Relation To Broader Scientific Literature:**

The paper builds on prior work in imitation learning and model-based RL, contributing a novel integration of world models for stability improvements.

**Theoretical Claims:**

The theoretical aspects appear well-grounded, and the method aligns with established principles in imitation learning and model-based reinforcement learning.

---

> ### Author Rebuttal · Authors · 2025-03-30
>
> We sincerely thank you for your review and your recognition of its strengths, including its clarity, the simplicity and effectiveness of our approach, and the solid theoretical grounding with comprehensive empirical validation. Below, we provide detailed responses to your comments.
>
> ### 1: Rationale for Using Planning Instead of Reinforcement Learning
>
> The rationale behind using planning instead of reinforcement learning (RL) for sampling interactions needs further elaboration. Why was planning chosen over RL for sampling environment interactions? How does this choice impact performance in different settings?
>
> **Answer:**
> Planning is particularly effective for complex tasks with high-dimensional observation and action spaces, such as the Dog Run task. On simpler tasks with low-dimensional observation and action spaces, like the Walker Run task, the performance difference between planning and directly using the policy prior for action sampling is not as significant. Moreover, the planning process is initialized with the RL policy prior, which means it generally enhances performance.
>
> ---
>
> ### 2: Discussion of Failure Cases and Limitations
>
> A deeper discussion of failure cases and limitations would strengthen the contribution.
>
> **Answer:**
> Thank you for the suggestion. We have discussed the potential instability caused by an imbalance between the policy and critic in the paragraph titled Balancing Critic and Policy Training on Page 5 of our manuscript. While we mitigate this issue using gradient penalty techniques, which help achieve strong performance across various tasks, the potential for instability may persist in other settings. We will provide a more detailed explanation and additional examples in the revised manuscript.
>
> ---
>
> ### 3: Generalization to Unseen Tasks or Domains
>
> Could the proposed method generalize to unseen tasks or domains beyond the evaluated environments?
>
> **Answer:**
> TD-MPC2 [1] has demonstrated the capability to generalize to unseen tasks after multi-task training. However, we did not conduct multi-task training for our IQ-MPC method. As a result, its generalization to unseen tasks would be limited. Expanding our approach to enable better generalization through multi-task training is an interesting direction for future research.
>
> ---
>
> ### 4: Code Availability and Reproducibility
>
> The lack of a code appendix makes it difficult to assess the reproducibility of the results. Do the authors plan to release the code to improve result reproducibility?
>
> **Answer:**
> To ensure reproducibility, we have provided our code in the following anonymous repository for review: https://anonymous.4open.science/r/reward-free-C1D5/. We also plan to make our code publicly available in the future.
>
> ---
>
> ### 5: Additional Related Work
>
> To the best of my knowledge, the related work has been discussed to a sufficient extent. However, the authors may want to consider Altmann et al. (2024), *"Discriminative Reward Co-Training"* as additional related work.
>
> **Answer:**
> Thank you for the suggestion. We will incorporate this additional related work in the revised manuscript.
>
>
> [1] Hansen, N., Su, H., & Wang, X. (2023). TD-MPC2: Scalable, Robust World Models for Continuous Control. *arXiv preprint arXiv:2310.16828.*

---

### Official Review · Reviewer_R4Uu · 2025-03-14

**Overall Recommendation:** 3

**Summary:**

The paper proposes a reward-free world model approach, IQ-MPC, for online imitation learning, addressing challenges in high-dimensional and complex tasks. The method integrates decoder-free latent dynamics models with inverse soft-Q learning, eliminating explicit reward modeling. By reformulating the optimization in Q-policy space and leveraging model predictive control (MPC), the approach achieves stable, expert-level performance on benchmarks including DMControl, MyoSuite, and ManiSkill2.

**Claims And Evidence:**

1. IQ-MPC outperforms baselines in tasks with high-dimensional observations and complex dynamics and mitigates instability during training. The compelling experimental results across locomotion and manipulation tasks over other baselines support this claim.

2. The reward-free world model learns effectively from expert demonstrations without explicit rewards, and can decode reliable rewards from the learned Q-values. The theoretical bijection between Q value and reward spaces is established. Moreover, Figure 16 and Table 7 show positive correlations between decoded and ground-truth rewards.

**Essential References Not Discussed:**

I don't think there are any essential references missing, but I'm not completely sure. I would still recommend considering the opinions of other reviewers.

**Experimental Designs Or Analyses:**

The experimental design is generally sound. The diverse task set and ablation on expert trajectory numbers  enhance the evaluation’s robustness.

However, the evaluated scope on visual tasks is limited (only 3 tasks). The noisy dynamics test (Figure 14) is limited to one task and minor noise with no comparison to baselines, may be insufficient to validate robustness claims.

**Methods And Evaluation Criteria:**

Methods: The integration of latent dynamics with inverse soft-Q learning is novel, and the gradient-free MPC planning aligns with recent trends (e.g., TD-MPC).

Evaluation: The benchmarks are diverse, covering low- and high-dimensional, state-based and visual-based, locomotion and manipulation tasks, which aligns with the problem of complex IL. The use of multiple baselines provides a fair comparison. However, The evaluation of noisy dynamics is limited to one task (Walker Run) with minor perturbations and lack comparison with other baselines, insufficient for a robust assessment.

**Other Comments Or Suggestions:**

The limitation statement of the proposed method is not found in the paper. It is recommended that the authors explicitly discuss the limitations of the proposed approach.

**Other Strengths And Weaknesses:**

Strengths:
1. Novel integration of inverse soft-Q learning with reward-free, decoder-free world models.
2. Comprehensive evaluation and analysis.
3. The paper is well-structured and explains its methodology clearly.

Weakness:
1. MPC’s computation overhead (Appendix F) is acknowledged but not fully quantified against baselines.
2. Robustness claim on noisy setting needs more experimental backing.

**Questions For Authors:**

1. Why do the authors use such a large number of expert trajectories in the experiment (e.g., 100 expert trajectories for visual tasks in DMC, totaling 100 × 500 expert transition steps)? Given that the total number of online training steps is 1M and the common choice in this setting is just 10 expert trajectories [1][2], this seems unusually high. Although Figure 5 demonstrates the method’s robustness to the number of expert trajectories, showing that performance remains solid with just 10 expert trajectories, this ablation study is only conducted on two tasks. I am curious about the reasoning behind the initial choice.
2. Do the authors plan to extend the proposed method to real-robot experiments? I am eager to see the results in real-world settings, as the paper only presents experiments in simulated environments.

[1] Rafailov R, Yu T, Rajeswaran A, et al. Visual adversarial imitation learning using variational models[J]. Advances in Neural Information Processing Systems, 2021, 34: 3016-3028.

[2] Wan S, Wang Y, Shao M, et al. Semail: eliminating distractors in visual imitation via separated models[C]//International Conference on Machine Learning. PMLR, 2023: 35426-35443.

**Relation To Broader Scientific Literature:**

The key contributions of the paper are closely related to imitation learning, model-based rl and inverse rl, particularly in the areas of decoder-free world models and inverse soft Q-learning.

**Theoretical Claims:**

The paper includes theoretical analysis in Section 4.2 and Appendix H, focusing on the learning objective’s suboptimality bound (Lemma 4.1) and consistency loss (Appendix H.3). Lemma 4.1 bounds the suboptimality of the policy based on distribution discrepancy between agent and expert and dynamics modeling error. The proof in Appendix H is mathematically sound, aligning with standard RL theory. Appendix H.3 links consistency loss to minimizing KL divergence between true and learned dynamics under Gaussian assumptions, but this analysis seems relatively trivial.

---

> ### Author Rebuttal · Authors · 2025-03-30
>
> We sincerely thank you for your review and your recognition of its strengths, including our novel approach, comprehensive evaluation, and clear presentation. Below, we provide detailed responses to your comments.
>
> ### 1: Limited Scope on Visual Tasks and Noisy Dynamics Test
>
> The evaluated scope on visual tasks is limited (only 3 tasks). The noisy dynamics test (Figure 14) is restricted to one task and minor noise with no comparison to baselines, which may be insufficient to validate robustness claims.
>
> **Answer:**
> We have conducted additional experiments with visual observations on the Quadruped Walk task, as well as noisy environment dynamics experiments on the Cheetah Run and the dexterous hand manipulation task Key Turn. The table below compares the performance degradation in terms of success rate under noisy environment dynamics between our approach and IQL+SAC:
>
> | $p_{tremble}$ | IQL+SAC        | HyPE         | IQ-MPC         |
> |----------------|----------------|--------------|----------------|
> | 0              | 0.72 ± 0.04   | 0.55 ± 0.09 | 0.87 ± 0.03   |
> | 0.005          | 0.54 ± 0.13   | 0.43 ± 0.14 | 0.79 ± 0.07   |
> | 0.01           | 0.28 ± 0.16   | 0.33 ± 0.13 | 0.61 ± 0.12   |
>
> The results demonstrate that our approach is more robust to noisy environment dynamics compared to IQL+SAC and HyPE. Additional experimental results for Quadruped Walk with visual observations and Cheetah Run with noisy dynamics are available in the anonymous repository: https://anonymous.4open.science/r/reward-free-C1D5/.
>
> Regarding concerns about the noise level, our noisy setting aligns with HyPE [1], which applied $p_{tremble}$ values of 0.01 and 0.025 for locomotion tasks, comparable to our noise levels in Figure 14. We hope the experimental results provided above sufficiently address your concerns.
>
> [1] Ren, J., Swamy, G., Wu, Z. S., Bagnell, J. A., & Choudhury, S. (2024). Hybrid Inverse Reinforcement Learning. *arXiv preprint arXiv:2402.08848.*
>
> ---
>
> ### 2: MPC Computation Overhead
>
> MPC’s computation overhead (Appendix F) is acknowledged but not fully quantified against baselines.
>
> **Answer:**
> We have provided a comparative analysis of computational overhead between our approach (both with and without MPC implementation) and other baseline methods (IQL+SAC, CFIL+SAC, HyPE, HyPER) in Figure 15 of our manuscript. Should any further clarification or additional comparative data be required to support your review, we would be happy to provide more detailed explanations or conduct supplementary analyses. Please feel free to share any specific aspects you would like us to elaborate on, and we will gladly address them accordingly.
>
> ---
>
> ### 3: Number of Expert Trajectories
>
> Why do the authors use such a large number of expert trajectories (e.g., 100 expert trajectories for visual tasks in DMC, totaling 100 × 500 expert transition steps)? Given the total number of online training steps is 1M and the common choice in this setting is just 10 expert trajectories [1][2], this seems unusually high. While Figure 5 shows the method’s robustness to fewer expert trajectories, this ablation is only conducted on two tasks. What is the reasoning behind the initial choice?
>
> **Answer:**
> Our approach can also effectively learn using only 10 expert trajectories across all three visual tasks, consistent with the settings in [1][2]. The results using 10 expert trajectories are available in our anonymous repository: https://anonymous.4open.science/r/reward-free-C1D5/. We initially chose to use 100 demonstrations to maintain consistency in the number of expert trajectories across tasks of varying difficulty, as using only 10 expert demonstrations for all tasks would be challenging.
>
> ---
>
> ### 4: Real-World Experiments
>
> Do the authors plan to extend the proposed method to real-robot experiments? I am eager to see the results in real-world settings, as the paper only presents experiments in simulated environments.
>
> **Answer:**
> Yes, we plan to extend our method to real-world experiments in future work. Given our method's promising performance in simulation, we believe it has strong potential to tackle real-world robotic tasks. We appreciate your interest and look forward to sharing future results.

---

> > ### Comment · Reviewer_R4Uu · 2025-04-09
> >
> > The authors' response addressed most of my concerns. I will maintain my positive score.

---

### Decision · Program_Chairs · 2025-05-01

**Decision:**

Accept (poster)

**Comment:**

The AC read the paper, reviews and discussions. 3/4 reviewers suggested weak accept, which the AC agrees with. 1/4 reviewer suggested accept. The content is ok in terms of addressing challenges in high-dimensional and complex tasks, with MPC based algorithm. The tasks are not very challenging, not sure if the Dreamer line of algorithms can also do it for free.